# RevealLayer: Disentangling Hidden and Visible Layers via Occlusion-Aware Image Decomposition

**Binhao Wang** [1 2 *]  **Shihao Zhao** [1 2 *]  **Bo Cheng** [2 * †]  **Qiuyu Ji** [1 2]  **Yuhang Ma** [2]
**Liebucha Wu** [2]  **Shanyuan Liu** [2]  **Dawei Leng** [2 ‡]  **Yuhui Yin** [2]

## Abstract

Recent diffusion-based approaches have made substantial progress in image layer decomposition. However, accurately decomposing complex natural images remains challenging due to difficulties in occlusion completion, robust layer disentanglement, and precise foreground boundaries. Moreover, the scarcity of high-quality multi-layer natural image datasets limits advancement. To address these challenges, we propose **RevealLayer**, a diffusion-based framework that decomposes an RGB image into multiple RGBA layers, enabling precise layer separation and reliable recovery of occluded content in natural images. RevealLayer incorporates three key components: (1) a **Region-Aware Attention** module to disentangle hidden and visible layers; (2) an **Occlusion-Guided Adapter** to leverage contextual information to enhance overlapping regions; and (3) a **composite loss** to enforce sharp alpha boundaries and suppress residual artifacts. To support training and evaluation, we introduce **RevealLayer-100K**, a high-quality multi-layer natural image constructed through a collaboration between automated algorithms and human annotation, and further establish **RevealLayerBench** for benchmarking layer decomposition in general natural scenes. Extensive experiments demonstrate that RevealLayer consistently outperforms existing approaches in layer decomposition.

## 1. Introduction

Recent text-to-image diffusion models (Esser et al., 2024) have achieved remarkable progress in image quality and diversity. However, they are primarily designed for single-layer RGB generation, leaving multi-layer image modeling largely unexplored. Decomposing an RGB image into a background layer and multiple transparent RGBA foreground layers requires the model to handle complex occlusions, capture object hierarchy, and complete missing content, while maintaining consistency across visible regions. Achieving such decomposition in complex natural images remains a significant challenge due to the absence of explicit layer structure modeling in existing frameworks.

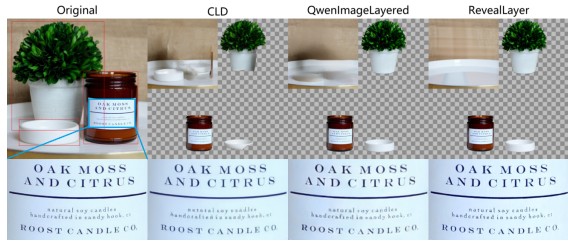

*Figure 1.* Qualitative comparison of layered image decomposition in natural scenes. RevealLayer exhibits strong capability in artifact removal, occlusion completion, and content consistency.

Most prior approaches tackle this problem via cascaded pipelines that sequentially perform instance segmentation, alpha matting, and image inpainting using specialized models. Such multi-stage designs are highly sensitive to intermediate errors, causing accumulated artifacts and degraded layer consistency, particularly in heavily occluded regions. More recently, end-to-end diffusion-based frameworks, such as LayerD (Suzuki et al., 2025) and OmniPSD (Liu et al., 2025a), jointly predict multiple layers to mitigate error accumulation. However, these methods typically lack explicit user guidance, offering limited control over layer semantics and ordering. Qwen-Image-Layered (Team, 2025a) introduces variable-layer decomposition, yet the number, order, and semantic meaning of the generated layers remain ambiguous. CLD (Liu et al., 2025b) uses bounding-box conditioning for controllable decomposition, but it is mostly restricted to stylized poster images and tends to produce residual artifacts and blurred object edges. Consequently, existing approaches remain insufficient for controllable, occlusion-aware layer decomposition in real-world

---

[*]Equal contribution [†] Project Lead. [‡] Corresponding Author. [1]Wenzhou University [2]360 AI Research. Correspondence to: Dawei Leng <lengdawei@360.cn>.

*Proceedings of the 43rd International Conference on Machine Learning*, Seoul, South Korea. PMLR 306, 2026. Copyright 2026 by the author(s).

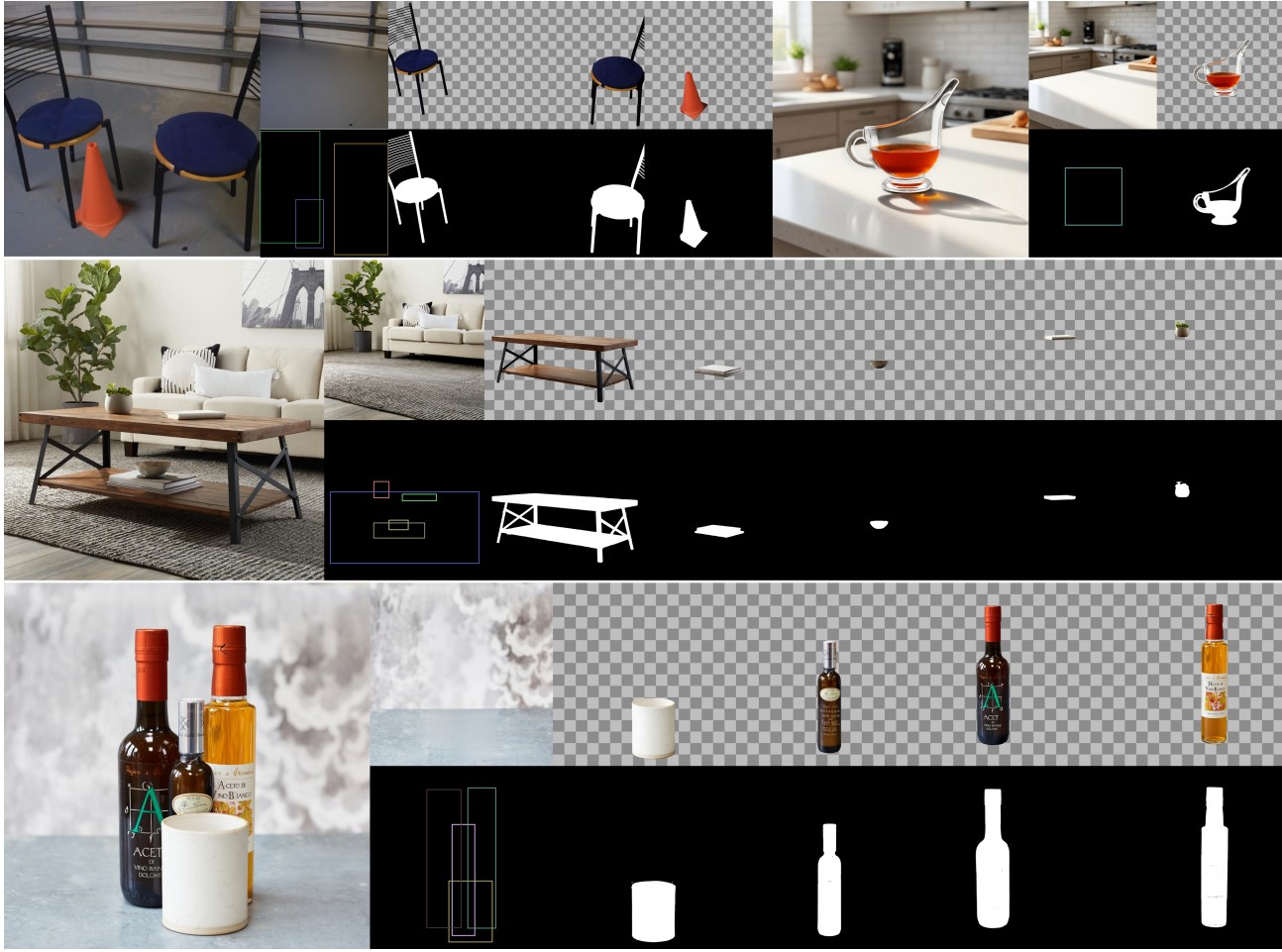

*Figure 2.* RevealLayer decomposes an input image into multiple RGBA layers with explicit transparency according to user-specified bounding boxes. Our method demonstrates strong capability in completing overlapping regions, accurately recovering object boundaries, and handling transparent objects, while also maintaining high visual consistency in the visible regions.

scenes, motivating the development of more flexible and robust multi-layer decomposition methods.

We observe that region-level layer disentanglement and intermediate feature enhancement play a critical role in improving both layer decomposition and occlusion completion. Based on this insight, we propose **RevealLayer**, a diffusion-based framework that decomposes an image into multiple RGBA layers under user-specified bounding-box guidance. RevealLayer integrates three modules: (1) a Region-Aware Attention for region-level separation of visible and hidden content, (2) an Occlusion-Guided Adapter for occlusion-aware reconstruction, and (3) a composite loss (alpha + orthogonality) to enforce sharp boundaries and avoid layer ambiguity. Figure 1 shows that Qwen-Image-Layered and CLD suffer from background artifacts and incomplete foregrounds, whereas RevealLayer effectively suppresses target-related artifacts, completes occluded regions, and preserves consistency in visible regions.

Progress in natural image layer decomposition is limited by the lack of large-scale, high-quality multi-layer datasets. To address this, we develop a comprehensive data pipeline and introduce **RevealLayer-100K**, a large-scale dataset of natural-scene images with high-quality RGBA layer annotations, along with **RevealLayerBench**, a benchmark for systematic evaluation on complex multi-layer layouts.

Our main contributions are summarized as following:

- We propose **RevealLayer**, a diffusion-based framework for controllable, bounding-box-guided decomposition of natural images into RGBA layers.

- We propose an **occlusion-aware paradigm** to disentangle visible and hidden content and recover occluded regions, consisting of Region-Aware Attention, Occlusion-Guided Adapter, and a composite loss.

- We develop a comprehensive data pipeline and introduce **RevealLayer-100K**, a large-scale dataset for natural image layer decomposition, together with **RevealLayerBench** for systematic evaluation.

- Experiments demonstrate that RevealLayer achieves excellent performance in layer disentanglement, occluded content completion, and background fidelity.

**Conflict of Interest Disclosure.** The authors declare no financial conflicts of interest related to this work.

## 2. Related Work

### 2.1. Object Removal and Image Matting

Object removal aims to remove specified objects and plausibly fill the resulting regions. PowerPaint (Zhuang et al., 2024) and SmartEraser (Jiang et al., 2025) combine segmentation with context-aware inpainting but often produce structural artifacts or unexpected content. RORem (Li et al., 2025a) and AttentiveEraser (Sun et al., 2025) leverage attention to model global context, yet struggle with semantic consistency in complex layouts. ObjectClear (Zhao et al., 2026) incorporates multi-scale, region-guided refinement, but remains limited in preserving background consistency and handling overlapping objects. Overall, existing methods still struggle in complex multi-object scenes with dense occlusions.

Image matting aims to recover accurate alpha mattes for foreground–background separation. Recent methods, such as SAM (Kirillov et al., 2023), leverage prompt-driven segmentation to guide matting but often produce imprecise boundaries and handle limited transparency. Matting Anything Models (MAM) (Li et al., 2024) leverages SAM to predict alpha mattes, it remains challenged by complex, overlapping object segmentation. Existing approaches remain insufficient for robust and controllable matting in complex natural images.

### 2.2. Image Composition and Object Insertion

Image composition and object insertion aim to place user-specified foreground objects into target scenes while preserving identity, spatial consistency, and visual harmony. Recent diffusion-based methods, such as AnyDoor (Chen et al., 2024) and Insert Anything (Song et al., 2026), have explored zero-shot object-level customization and reference-based insertion with spatial or textual control, while layout-controllable generation methods such as HiCo (Cheng et al., 2024) further improve spatial controllability from layout conditions. Related study (Lu et al., 2025) further investigates physically plausible composition with generative priors, e.g., handling lighting, shadows, and reflections. Meanwhile, efficient generative models (Ma et al., 2025) continue to improve the efficiency of image synthesis. Despite these advances, achieving precise region-level control in complex natural scenes remains challenging.

### 2.3. Image Layer Decomposition

Existing image decomposition methods either follow multi-stage cascaded pipelines or end-to-end frameworks. Cascaded approaches sequentially perform segmentation, alpha matting, and inpainting but suffer from error accumulation, resulting in inconsistent layers. End-to-end methods avoid such accumulation but struggle with fine-grained control over layer semantics and occlusion completion. For instance, LayerD (Suzuki et al., 2025) and OmniPSD (Liu et al., 2025a) lack explicit guidance, limiting control over layer order and semantics; Qwen-Image-Layered (Team, 2025a) introduces a variable-layer decomposition strategy, but the order and semantic meaning of the generated layers remain ambiguous; CLD (Liu et al., 2025b) uses bounding-box conditioning but mainly handles stylized poster images, with limited generalization and noticeable residual artifacts. These limitations motivate the need for RevealLayer, which enables controllable and occlusion-aware multi-layer decomposition in complex natural scenes.

## 3. Method

### 3.1. Problem Formulation

We formulate the task as a controllable layer decomposition problem. The objective is to decompose an input image into a background and a sequence of foreground layers, conditioned on user-specified layout boxes. Formally, given an input image $I \in \mathbb{R}^{H \times W \times 3}$ and a set of bounding boxes $\mathcal{B} = \{b_1, \ldots, b_N\}$ indicating the target objects, we aim to train a model $\mathcal{M}$ capable of predicting the disentangled layer set $\mathbf{I}_{bg}, \mathbf{I}_{\mathrm{fg}}^1, \ldots, \mathbf{I}_{\mathrm{fg}}^N \in \mathbb{R}^{H \times W \times 4}$:

$$\mathbf{I}_{bg}, \mathbf{I}_{\mathrm{fg}}^1 \ldots, \mathbf{I}_{\mathrm{fg}}^N = \mathcal{M}(I, \mathcal{B}) \tag{1}$$

The decomposition provides occlusion completion and layer consistency, explicitly eliminating object residual artifacts, and enabling fine-grained, controllable layer separation.

**RGBA-VAE** To accurately model transparency and compositional relationships in layered images, we adopt the Multi-Layer Transparent Image Autoencoder (TransVAE) from ART (Pu et al., 2025), a unified variational autoencoder for RGBA images. Since TransVAE is originally trained on graphic design data, we fine-tune it on natural images to bridge the domain gap, and use the adapted model as the image autoencoder in RevealLayer. We compute $\hat{\mathbf{I}}_{\mathrm{fg}}^i = (0.5\mathbf{I}_{\mathrm{fg},\alpha}^i + 0.5) \times \mathbf{I}_{\mathrm{fg,RGB}}^i$, converting the transparent-background image $\mathbf{I}_{\mathrm{fg}}^i$ into a gray-background image $\hat{\mathbf{I}}_{\mathrm{fg}}^i$.

### 3.2. RevealLayer-DiT

In this section, we present RevealLayer, a controllable layer decomposition framework built upon the FLUX.1 [dev] (Black Forest Labs, 2024) architecture, which adopts

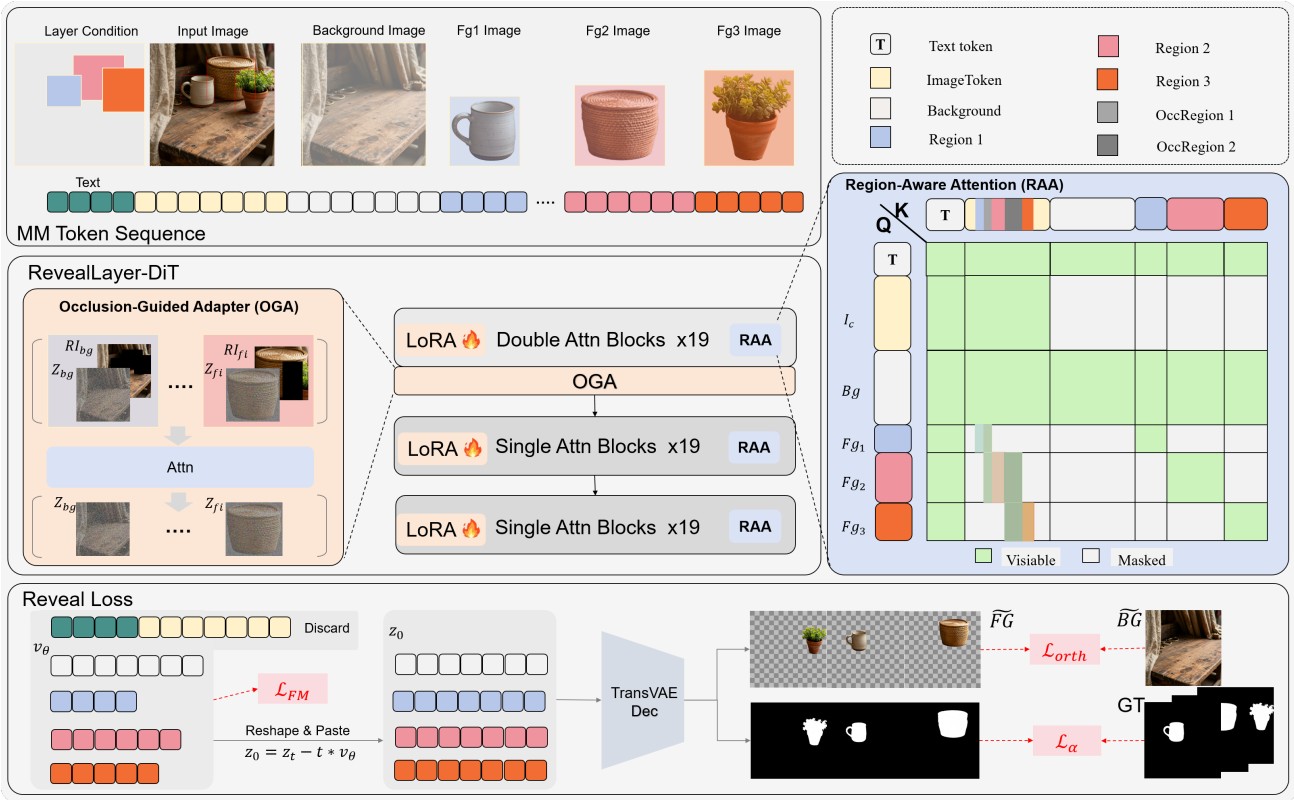

**Figure 3.** The framework of RevealLayer, a controllable layer decomposition architecture based on FLUX. It incorporates Region-Aware Attention (RAA) and an Occlusion-Guided Adapter (OGA) to enhance layer disentanglement and occlusion completion, while alpha and orthogonality losses are employed to suppress boundary blur and residual artifacts.

the MM-DiT as its backbone. We formulate the decomposition task as a variable-length sequence modeling problem. The input image $\mathbf{I}$, the background image $\mathbf{I}_{bg}$, and all the foreground image layers $\{\hat{\mathbf{I}}_{fg}^i\}_{i=1}^N$ are fed into the VAE encoder $\mathcal{E}_{VAE}$ to extract latent representations, and then crop and flattened into latent tokens with different lengths:

$$\mathbf{z}_0^c = \mathsf{Flatten}(\mathcal{E}_{VAE}(\mathbf{I})), \mathbf{z}_0^0 = \mathsf{Flatten}(\mathcal{E}_{VAE}(\mathbf{I}_{bg})), \quad (2)$$

$$\mathbf{z}_0^i = \mathsf{Flatten}(\mathsf{Crop}(\mathcal{E}_{VAE}(\hat{\mathbf{I}}_{fg}^i), b_i)), \quad i = 1, \cdots, N \quad (3)$$

Finally, the multi-layer image latent is represented as a unified token sequence $S$, formed by concatenating tokens of varying lengths.

$$z_0 = [z_0^c; z_0^0; z_0^1; \ldots; z_0^N] \in \mathbb{R}^{L \times D} \quad (4)$$

To enable the model to distinguish and correlate layers, we integrate 3D Rotary Positional Embeddings (3D-RoPE) (Pu et al., 2025) into $z$. According to Rectified Flow, the intermediate state $z_t$ and velocity $v_t$ at timestep $t$ is defined as:

$$z_t = tz_0 + (1-t)z_1 \quad (5)$$

$$v_t = \frac{dz_t}{dt} = z_0 - z_1 \quad (6)$$

where $z_0 \sim \mathcal{N}(\mathbf{0}, I)$, $z_1 \sim q_{data}(\mathbf{z})$. We jointly model all layers and optimize the model by computing the flow matching loss on each layer's variable-length token sequence. Thus, the overall flow matching loss can be expressed as

$$\mathcal{L}_{FM} = \sum_{i=1}^N \mathbb{E}_{(x_0, x_1, t, c_{text}, z_c)} ||v_{\theta(x_t, t, c_{text}, z_c)}^i - v_t^i||^2 \quad (7)$$

where $t$ is the timestep, $i$ is the $i_{th}$ layer, $z_c$ is the latent sequence of the conditional image, $c_{text}$ is the text condition, using the fixed prompt: "Decompose the image into foreground and background".

To stabilize layer-wise disentanglement and improve occlusion completion, we introduce Region-Aware Attention and an Occlusion-Guided Adapter, together with alpha and orthogonality losses to suppress boundary blur and background residual artifacts.

### 3.3. Region-Aware Attention

Since tokens from different layers are jointly modeled as a variable-length sequence, standard self-attention operates uniformly over all tokens, implicitly assuming homogeneous interactions across layers. Although 3D-RoPE en-

codes spatial positions, it does not impose explicit token-level constraints to distinguish or isolate information from different layers, leaving the model susceptible to inter-layer feature interference.

To address this limitation, we introduce a Region-Aware Attention(**RAA**), as illustrated in Figure 3. The attention mask imposes a structural constraint on cross-region token interactions, promoting region-consistent attention while reducing inter-layer information leakage. As a result, mask-guided attention suppresses cross-layer feature mixing and enables robust layer-wise disentanglement.

$$\text{Attention}(Q, K, V) = \text{Softmax}\left(\frac{QK^T}{\sqrt{d}} + M\right) V \quad (8)$$

Here, the **RAA** mask $M \in \mathbb{R}^{L \times L}$ governs the interaction between query tokens $q$ and key tokens $k$.

$$M_{\text{RAA}}(q, k) = \begin{cases} 1 & \text{if } q \in \mathcal{T} \cup \mathcal{L}_0 \vee k \in \mathcal{T} \\ & \text{or } q \in \mathcal{L}_i \wedge k \in \mathcal{L}_i \cup \mathcal{R}_i \\ 0 & \text{otherwise} \end{cases} \quad (9)$$

Formally, let $\mathcal{T}$ and $\mathcal{I}$ denote the token sets of the text prompt and the global image, respectively. For each decomposed layer, $\mathcal{L}_i$ represents the corresponding layer-specific tokens. To incorporate spatially aligned context, we define $\mathcal{R}_i \subset \mathcal{I}$ as the subset of global image tokens that spatially correspond to the bounding box $\mathcal{B}_i$. The attention mask is formulated as shown in Eq. (9).

RAA is designed to facilitate layer-wise representation disentanglement by prioritizing attention to region-consistent context. It particularly emphasizes spatially overlapping areas across layers while reducing interference from irrelevant layer information.

### 3.4. Occlusion-Guided Adapter

Although the RevealLayer-DiT backbone performs layer-wise content isolation via RAA, generating visually coherent content within overlapping regions remains challenging. To address this, we propose the Occlusion-Guided Adapter(**OGA**), which enhances semantic coherence in each layer's latent representation by incorporating localized information from the original image.

As shown in Figure 3, after DoubleAttnBlock the OGA module concatenates the conditional image features $z_c$ with the per-layer latent $\mathbf{z}_t^i$ along the channel dimension, and applies self-attention to enhance the semantic coherence, yielding the updated latent $\hat{\mathbf{z}}_t^i$.

$$\hat{z}_t^i = \text{Attn}\left(\mathbf{z}_t^i, z_c \odot M_i\right) \quad (10)$$

We formally define the layer-specific mask $M_i$ via set oper-

ations:

$$M_i = \begin{cases} 1 - \bigcup_{j=1}^{N} \mathcal{B}_j & \text{if } i = 0 \\ \mathcal{B}_i \cap \left(1 - \bigcup_{j \neq i} \mathcal{B}_j\right) & \text{if } i \geq 1 \end{cases} \quad (11)$$

Additionally, we employ an attention mask to enforce visibility among tokens within the same layer while preventing interactions between tokens from different layers.

### 3.5. Training Objectives

Our model is trained using a composite objective function that ensures both the fidelity of the generative process and the precision of the decomposed layers.

**Hard-Constraint Alpha Loss.** While the Flow Matching loss guarantees generative stability, layer decomposition demands pixel-level boundary precision. To align the generated the boundary and transparency with the ground truth, we propose the Hard-Constraint Alpha Loss ($\mathcal{L}_\alpha$), which applies a focal-style penalty to refine foreground generation. Specifically, we first estimate the clean latent $\hat{z}_0$ from the current noisy state $z_t$ and the predicted velocity $v_\theta(t)$, and then decode it via the TranspVAE decoder $\mathcal{D}$ to obtain $\hat{I}_{RGBA}^i$, which formed by alpha map $\hat{I}_\alpha^i$ and $\hat{I}_{RGB}^i$.

$$\hat{z}_0 = z_t - t \cdot v_\theta(z_t, t, c_{text}) \quad (12)$$
$$\mathcal{D}(\hat{z}_0) = \left(\hat{I}_\alpha^i, \hat{I}_{RGB}^i\right)$$

$$\delta_i = \tau \cdot |\hat{I}_\alpha^i - \hat{I}_{\alpha, gt}^i| \quad (13)$$

where $\hat{I}_{\alpha, gt}^i$ denotes the ground-truth alpha channel, and $\tau = 0.95$ is the scaling factor, $\delta_i$ is the pixel-wise mse loss on the alpha channel of the $i_{th}$ foreground layer. To strictly constrain complex transition regions, we formulate the loss by aggregating penalties across all decomposed layers. The hard constraint alpha loss is defined as

$$\mathcal{L}_\alpha = -\sum_{i=1}^{N} \left((\delta_i)^\gamma \cdot \log(1 - \delta_i + \epsilon)\right) \quad (14)$$

where $\epsilon$ is a small constant for numerical stability, $\gamma$ is 1.5. To address the challenge of blurred object boundaries, we employ a focal-loss–style alpha supervision. This log-based loss focuses on hard boundary pixels, encouraging sharper and more precise edges.

**Soft-Constraint Orthogonality Loss.** During contextual completion in overlapping background regions, target-related artifacts and residuals are likely to appear. To mitigate these effects, we introduce a soft-constrained orthogonality loss ($\mathcal{L}_{orth}$) in the pixel space to suppress undesired inter-layer interactions. Cosine-based orthogonality loss discourages low-frequency, structurally correlated foreground residuals in background reconstruction.

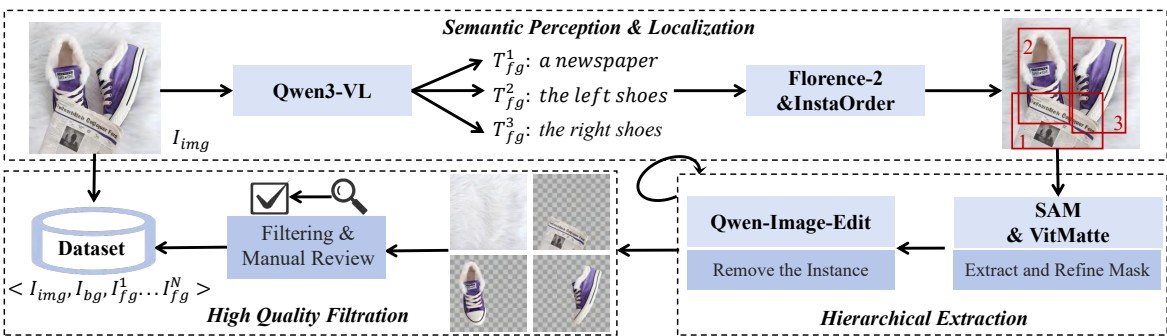

*Figure 4.* Dataset curation pipeline of RevealLayer-100K and RevealLayerBench.

$$\mathcal{L}_{orth} = \sum_{j=1}^{N} |\langle \hat{I}_{RGB}^{bg}, \hat{I}_{RGB}^{fg_j} \rangle_{R_i} - \langle I_{RGB}^{bg}, I_{RGB}^{fg_j} \rangle_{R_i}| \quad (15)$$

where $\langle \cdot, \cdot \rangle$ denotes the pixel-wise cosine similarity. $R_i$ denotes the mask corresponding to the region $\mathcal{B}_i$. $\hat{I}_{RGB}^{bg}$, $\hat{I}_{RGB}^{fg_j}$ are obtained via Eq. (12).

**Total Loss.** The final training objective is defined by the following loss function:

$$\mathcal{L} = \mathcal{L}_{FM} + \lambda_\alpha \mathcal{L}_\alpha + \lambda_o \mathcal{L}_{orth} \quad (16)$$

### 3.6. Dataset Construction

Existing open source multi-layer transparent datasets like MULAN (Tudosiu et al., 2024), Crello 20K (Yamaguchi, 2021), and PrismLayersPro 20K (Chen et al., 2025) are limited in scale and restricted to specific scenarios. Consequently, they lack complex natural scenes, as well as realistic environmental effects like shadows and reflections. Therefore, we introduce **RevealLayer-100K**, a large-scale multi-layer transparent dataset for natural images, and **RevealLayerBench**, providing tuples $\{I_{\text{img}}, I_{\text{bg}}, \{I_{\text{fg}}^i\}_{i=1}^N, \{\text{Bbox}_i\}_{i=1}^N\}$.

Our data is sourced from LAION-2B (Schuhmann et al., 2022), GRIT-20M (Peng et al., 2023), and internal collections. We design a robust extraction-removal pipeline to process this raw data. Specifically, Qwen3-VL (Team, 2025b) is utilized to filter images and generate pseudo-labels, which assist Florence-2 (Xiao et al., 2024) in instance detection. After sorting the instances via InstaOrder (Lee & Park, 2022), we iteratively use SAM-H (Kirillov et al., 2023) and Qwen-Image-Edit (Team, 2025c) for layer extraction, refining the masks with ViTMatte (Yao et al., 2024).

Since object removal inevitably alters background regions and complex occlusions challenge inpainting, we apply a background consistency filter and conduct a manual review, where annotators rigorously verify the consistency of foreground and background layers with the original image and

the fidelity of occluded region recovery. The overall pipeline is illustrated in Figure 4. Additionally, two augmentation pipelines (detailed in the Appendix A) are designed to enhance the robustness of background completion regions and address the scarcity of occluded instances.

## 4. Experiment

### 4.1. Experiment Setting

**Datasets.** To support training and evaluation, we introduce **RevealLayer-100K**, a high-quality multi-layer natural image constructed through a collaboration between automated algorithms and human annotation, and further establish **RevealLayerBench** for benchmarking layer decomposition in natural scenes.

**Implementation Details.** Our method is built upon the Flux.1[dev] model, which we fine-tune using LoRA (Hu et al., 2022) with a rank of 64. The training process is managed by the Prodigy optimizer, configured with a learning rate of 1.0. We conduct the training for 50,000 iterations using a global batch size of 8. All input images are resized to a uniform resolution of 1024 on their long side. In Eq. (16), $\lambda_\alpha = 1.0$ and $\lambda_o = 1.0$.

### 4.2. Quantitative Result

We comprehensively evaluate our model on object removal (OBER-Test (Zhao et al., 2026), RevealLayerBench), matting (AIM-500 (Li et al., 2021), RefMatte-RW100 (Li et al., 2023)), and natural image multi-layer decomposition (RevealLayerBench) to assess background recovery, foreground accuracy, and the controllability of multi-layer disentanglement. Additional visual results are provided in the supplementary material.

**Object Removal.** We compare our layer-decomposition-based approach with state-of-the-art object removal methods on the background layer, including PowerPaint (Zhuang et al., 2024), SmartEraser (Jiang et al., 2025), RORem (Li et al., 2025a), AttentiveEraser (Sun et al., 2025), ObjectClear (Zhao et al., 2026), and OmniPaint (Yu et al., 2025).

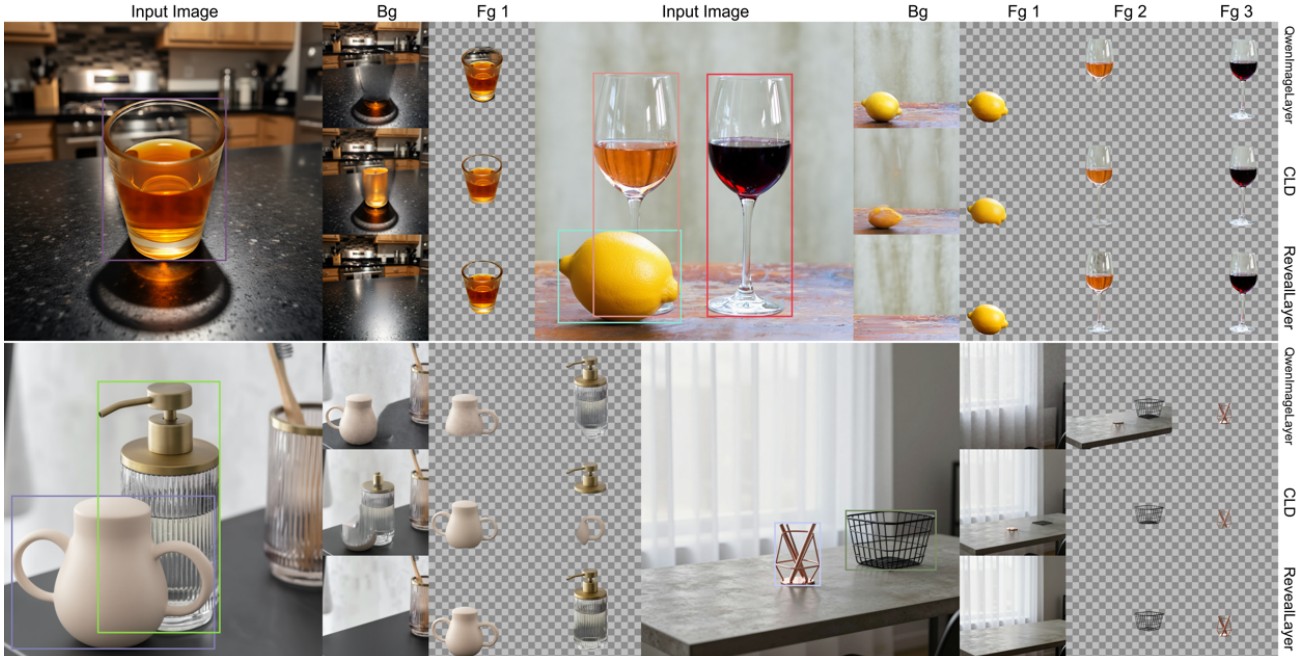

*Figure 5.* Qualitative comparison of Image-to-Multi-RGBA. Qwen-Image-Layered and CLD suffer from artifacts in overlapping regions, missing foreground objects, and poor consistency in visible areas, while RevealLayer demonstrates strong performance in layer disentanglement, occluded content recovery, and accurate object boundary reconstruction.

*Table 1.* Background Reconstruction Results for Object Removal on OBER-Test and RevealLayerBench. For ObjectClear∗, the reported metrics exclude the post-processing step involving background blending with the original image.

| Guidance | Method | Source | OBER-Test | | | | RevealLayerBench | | | |
|---|---|---|---|---|---|---|---|---|---|---|
| | | | PSNR↑ | SSIM↑ | LPIPS↓ | FID↓ | PSNR↑ | SSIM↑ | LPIPS↓ | FID↓ |
| Mask | PowerPaint (Zhuang et al., 2024) | ECCV'24 | 22.64 | 0.7297 | 0.1237 | 86.15 | 18.65 | 0.7674 | 0.2172 | 136.37 |
| | SmartEraser (Jiang et al., 2025) | CVPR'25 | 23.97 | 0.7312 | 0.1155 | 71.26 | 19.05 | 0.7288 | 0.2840 | 119.96 |
| | RORem (Li et al., 2025a) | CVPR'25 | 24.83 | 0.7578 | 0.1120 | 62.58 | 21.59 | 0.8197 | 0.1678 | 74.17 |
| | AttentiveEraser (Sun et al., 2025) | AAAI'25 | 26.40 | 0.7716 | 0.1251 | 54.06 | 22.15 | 0.8149 | 0.1774 | 80.41 |
| | ObjectClear∗ (Zhao et al., 2026) | CVPR'26 | 27.57 | 0.7671 | 0.0840 | 31.06 | 23.81 | 0.7812 | 0.2083 | 60.88 |
| | OmniPaint (Yu et al., 2025) | ICCV'25 | 29.05 | 0.8736 | **0.0521** | **20.70** | 24.83 | **0.8471** | 0.1374 | 54.22 |
| Box | RevealLayer | Ours | **30.16** | **0.9153** | 0.0694 | 25.62 | **25.53** | 0.8429 | 0.1483 | **53.81** |

*Table 2.* Quantitative Foreground Matting Results on AIM-500 and RefMatte-RW100.

| Method | Source | AIM500 | | | | | RefMatte-RW100 | | | | |
|---|---|---|---|---|---|---|---|---|---|---|---|
| | | MSE↓ | MAD↓ | SAD↓ | CONN↓ | SoftIoU↑ | MSE↓ | MAD↓ | SAD↓ | CONN↓ | SoftIoU↑ |
| SAM-H (Kirillov et al., 2023) | ICCV'23 | 0.0635 | 0.0705 | 117.13 | 25.34 | 0.8757 | 0.0697 | 0.0721 | 126.25 | 23.75 | 0.8690 |
| SAM2-L (Ravi et al., 2025) | ICLR'25 | 0.0720 | 0.0790 | 130.06 | 22.10 | 0.8597 | 0.0587 | 0.0611 | 107.21 | 16.10 | 0.8769 |
| SAM3 (Carion et al., 2025) | arXiv | 0.0510 | 0.0581 | 96.44 | **15.07** | 0.8555 | 0.0406 | 0.0430 | 72.21 | 11.51 | 0.9062 |
| MAM (Li et al., 2024) | CVPR'24 | 0.0168 | 0.0290 | 47.59 | 22.32 | 0.8775 | 0.0267 | 0.0369 | 62.13 | 20.26 | 0.8912 |
| RevealLayer | Ours | **0.0107** | **0.0195** | **32.39** | 17.78 | **0.9048** | **0.0108** | **0.0148** | **25.49** | **9.15** | **0.9592** |

Evaluations are conducted on two benchmarks: OBER-Test, which focuses on natural images with a single target object, and RevealLayerBench, which contains complex natural images with multiple objects and cluttered layouts.

As shown in Table 1, our bounding-box-guided layer decomposition method achieves strong background reconstruction performance on both single-object and complex multi-object datasets. PowerPaint and SmartEraser frequently produce

residual structural artifacts or hallucinated regions, while RORem and AttentiveEraser better adhere to mask constraints but still struggle to maintain global semantic consistency under complex spatial arrangements. In contrast, our approach achieves the highest PSNR and SSIM scores, reflecting improved pixel-level and structural fidelity for background reconstruction, and also achieves lower LPIPS and FID, indicating enhanced perceptual quality and overall

*Table 3.* Quantitative Evaluation of Multi-Layer Decomposition under Complex Multi-Object Layouts on RevealLayerBench. Performance metrics evaluated on 3-layer images of 1024×1024 resolution with default settings.

| Method | Background-level | | | Foreground-level | | | | Q-Insight | | | Performance | |
|---|---|---|---|---|---|---|---|---|---|---|---|---|
| | PSNR↑ | LPIPS↓ | FID↓ | PSNR↑ | LPIPS↓ | FID↓ | SoftIoU↑ | Consistency↑ | Fidelity↑ | Editability↑ | Time(s)↓ | Mem(G)↓ |
| Qwen-Image-Layered | 16.85 | 0.3293 | 142.01 | - | - | - | - | 4.00 | 3.88 | 4.05 | 418 | 76 |
| CLD | 19.75 | 0.2293 | 127.77 | 26.42 | 0.0433 | 43.64 | 0.8304 | 4.00 | 3.76 | 4.06 | **97** | 62 |
| Ours | **25.53** | **0.1483** | **53.81** | **32.13** | **0.0217** | **18.42** | **0.9432** | **4.09** | **3.91** | **4.14** | 122 | **60** |

*Table 4.* Quantitative Results of Ablation Studies. All ablations are conducted on RevealLayer-100K using identical training configurations and equal training budgets to assess the contribution of each proposed module.

| Variant | RAA | OGA | OrthLoss | AlphaLoss | Background-level | | | Foreground-level | | | |
|---|---|---|---|---|---|---|---|---|---|---|---|
| | | | | | PSNR↑ | LPIPS↓ | FID↓ | PSNR↑ | LPIPS↓ | FID↓ | SoftIoU↑ |
| (a) | ✗ | ✗ | ✗ | ✗ | 22.08 | 0.155 | 59.54 | 30.62 | 0.025 | 20.86 | 0.9224 |
| (b) | ✓ | ✗ | ✗ | ✗ | 23.80 | 0.161 | 58.12 | 31.68 | 0.023 | 19.66 | 0.9317 |
| (c) | ✓ | ✓ | ✗ | ✗ | 24.97 | 0.146 | 57.51 | 31.78 | 0.024 | 19.65 | 0.9271 |
| (d) | ✓ | ✓ | ✓ | ✗ | 24.96 | **0.142** | **53.63** | 32.09 | 0.022 | **18.16** | 0.9418 |
| (e) | ✓ | ✓ | ✗ | ✓ | 24.86 | 0.153 | 57.98 | 31.97 | 0.023 | 19.17 | 0.9420 |
| (f) | ✓ | ✓ | ✓ | ✓ | **25.53** | 0.148 | 53.81 | **32.13** | **0.021** | 18.42 | **0.9432** |

realism. In contrast, RevealLayer achieves the best PSNR on both datasets and obtains the best or second-best SSIM, LPIPS, and FID scores, demonstrating its effectiveness in reconstructing structurally faithful and perceptually plausible backgrounds. Notably, compared with OmniPaint, which is also based on FLUX.1[dev], our method achieves higher PSNR on both datasets, suggesting that the improvement is not merely due to the backbone but also benefits from our box-guided layer decomposition design. For more detailed quantitative results and visualizations on the test datasets, refer to the Appendix B.

**Object Matting.** We quantitatively evaluate our method on the alpha channel of foreground RGBA images using two benchmarks: AIM500 for natural image matting and RefMatte-RW100 for real-world portrait matting. The evaluation includes both general-purpose segmentation models (e.g., SAM3 (Carion et al., 2025)) and task-specific matting methods (e.g., MAM (Li et al., 2024)).

As shown in Table 2, our method demonstrates exceptional performance on both benchmarks. MAM achieves high pixel-level accuracy but often produces fragmented foregrounds, whereas SAM3 preserves semantic structure at the cost of larger alpha errors. Our generative decomposition approach achieves a balanced trade-off among fidelity, edge sharpness, and content consistency. It also demonstrates fine-grained alpha reconstruction on both datasets.

**MultiLayer Decomposition.** On the complex multi-object benchmark RevealLayerBench, we evaluate background and foreground disentanglement, generation quality, and foreground alpha accuracy. We further use Q-Insight (Li et al., 2025b) for zero-shot assessment of consistency, fidelity, and editability, leveraging MLLM reasoning and reinforcement

learning for interpretable evaluation.

As shown in Table 3, compared with Qwen-Image-Layered and CLD, our method achieves significant advantages on both background and foreground layers, demonstrating more accurate layer decomposition and stronger recovery of overlapping regions. It achieves higher soft IoU in the alpha channels of multiple foregrounds, particularly preserving fine-grained edge details. In terms of Q-Insight, our approach attains the highest Consistency, Fidelity, and Editability scores, highlighting superior structural integrity and manipulability of the decomposed layers. The RAA module slightly increases computation but provides a favorable efficiency–performance trade-off.

*Table 5.* The human evaluation results on **RevealLayerBench**. The numerical range is from 0 to 100, with a higher score indicating better performance.

| Method | LayersNums↑ | Bg Q↑ | Fg Q↑ |
|---|---|---|---|
| Qwen-Image-Layered | 57 | 20 | 52 |
| CLD | 96 | 13 | 40 |
| RevealLayer | **99** | **85** | **90** |

### 4.3. Qualitative Result

Figure 5 shows layer decomposition results on natural images. Unlike Qwen-Image-Layered and CLD, which suffer from background artifacts and incomplete foregrounds, RevealLayer delivers more accurate occluded region recovery, sharper boundaries, and consistent visible content. Quantitative evaluation using a multi-round, multi-participant protocol on the number of layers (**LayersNums**), background quality (**Bg Q**), and foreground completeness (**Fg Q**). Table 5 shows RevealLayer achieves the highest human

*Table 6.* Robustness analysis under different types of abnormal bounding-box inputs.

| Variant | Background-level | | | Foreground-level | | | |
|---|---|---|---|---|---|---|---|
| | PSNR↑ | LPIPS↓ | FID↓ | PSNR↑ | LPIPS↓ | FID↓ | SoftIoU↑ |
| Excessive 10%–20% | 25.30 | 0.1507 | 54.28 | 31.69 | 0.0253 | 18.77 | 0.9368 |
| Offset 0–5% | 25.30 | 0.1518 | 56.31 | 30.92 | 0.0260 | 20.22 | 0.9278 |
| Offset 5%–10% | 24.57 | 0.1615 | 61.36 | 27.22 | 0.0425 | 28.40 | 0.8569 |
| Inadequate 0–5% | 25.26 | 0.1515 | 56.06 | 31.55 | 0.0237 | 19.22 | 0.9373 |
| Inadequate 5%–10% | 24.64 | 0.1610 | 62.36 | 28.01 | 0.0396 | 28.96 | 0.8757 |
| Precise bbox(ours) | **25.53** | **0.1483** | **53.81** | **32.13** | **0.0217** | **18.42** | **0.9432** |

preference scores in layer controllability, as well as foreground and background quality. Detailed evaluation criteria is included in the supplementary material B.7.

### 4.4. Ablation Study

We conduct comprehensive ablation studies on RevealLayer-100K to systematically analyze the contribution of each proposed component. All variants are trained under identical settings with the same training budget to ensure a fair comparison. As shown in (a) and (b), the RAA module significantly improves layer disentanglement, as reflected by notable PSNR gains and FID reductions for both background and foreground. As shown in (b) and (c), the OGA module effectively leverages contextual information to enhance overlapping regions, yielding higher background PSNR and more stable perceptual metrics. As shown in (c), (d), and (e), the orthogonality and alpha losses respectively suppress residual inter-layer artifacts and enforce sharp alpha boundaries, leading to lower background FID and improved foreground LPIPS and Soft IoU. Combining all modules, the full model achieves the best overall performance, with each component complementing the others to enhance both background and foreground reconstruction.

### 4.5. Robustness to Bounding-Box Perturbations

Since RevealLayer uses bounding boxes as the primary user guidance, we further evaluate its robustness to inaccurate box inputs. As shown in Table 6, mild box perturbations only cause negligible performance degradation. For example, when the input box is moderately enlarged by 10%–20%, the background PSNR decreases slightly from 25.53 dB to 25.30 dB, while LPIPS and FID remain close to those obtained with precise boxes. Similarly, small spatial offsets within 5% and slightly inadequate boxes within 5% lead to only minor changes in both background reconstruction and foreground decomposition metrics.

In contrast, more severe perturbations, such as 5%–10% spatial offsets or 5%–10% inadequate boxes, result in more noticeable performance drops. This is expected because

severely shifted or incomplete boxes may exclude visible object regions or include excessive background regions, making foreground-background separation and occlusion completion more ambiguous. Nevertheless, the model maintains reasonable reconstruction quality under these challenging settings, demonstrating that RevealLayer is not overly sensitive to small annotation errors and can tolerate practical bounding-box inaccuracies.

## 5. Conclusion

In this paper, we propose RevealLayer, a framework for layered image decomposition, decoupling RGB images into coherent background and RGBA foregrounds using only instance bounding boxes. Leveraging region-aware attention, an occlusion-guided adapter, and a combined loss, our method robustly handles complex natural scenes. We also introduce RevealLayer-100K, a large-scale multi-layer occlusion dataset for natural images. Extensive experiments show state-of-the-art performance, and the model generalizes to downstream tasks such as object removal, matting, inpainting, and other layer-based image editing applications. Nevertheless, challenging cases may still arise under severely inaccurate bounding boxes, heavy occlusion, transparent or translucent regions, and dense repetitive textures, where layer separation and occlusion completion become inherently ambiguous. Future work will focus on performance optimization, robustness improvement, and enabling multi-round interactive editing and generation.

## Impact Statement

This paper presents work whose goal is to advance the field of Machine Learning. There are many potential societal consequences of our work, none which we feel must be specifically highlighted here.

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

# A. Dataset Construction Details

This section provides a detailed methodology for constructing our dataset, which comprises a large-scale training set, **RevealLayer-100K**, and a high-quality evaluation set, **RevealLayerBench**. We begin by curating our initial data pool from diverse sources, including LAION-2B (Schuhmann et al., 2022), GRIT-20M (Peng et al., 2023), and internal collections.

**Extraction Pipeline (Pipeline 1).** We leverage the Qwen3-VL-30B-A3B (Team, 2025b) model to filter out images with simplistic or cluttered backgrounds and to generate textual descriptions for key instances within the retained images. These descriptions guide the Florence-2 model (Xiao et al., 2024) in performing text-conditioned instance detection, yielding precise bounding boxes. To ensure decomposition efficiency, images yielding more than eight instances are discarded. For each image, we employ a combination of SAM-H (Kirillov et al., 2023) and Qwen-Image-Edit (Team, 2025c) to generate coarse masks and inpaint the removed area, which are subsequently refined by VitMatte (Yao et al., 2024) to produce high-quality RGBA foreground layers $\{I_{\text{fg}}^i\}_{i=1}^N$. The remaining region constitutes the background layer, $I_{\text{bg}}$, which is derived via object removal. Notably, the extraction sequence is determined by the InstaOrder model (Lee & Park, 2022) to preserve coherent layer ordering. Since $I_{\text{bg}}$ in this pipeline is a reconstruction without a ground truth, quality assurance relies primarily on perceptual image filters and manual review.

**Generative Pipeline (Pipeline 2).** To enhance the diversity and robustness of our dataset, we introduce a complementary synthesis pipeline. Unlike Pipeline 1, where backgrounds are derived via removal, this approach generates backgrounds from scratch, effectively providing a pristine ground truth. Specifically, we utilize QwenVL to generate textual descriptions for synthetic backgrounds, which are then rendered by the Zimage (Team, 2025d) model to create $I_{\text{bg}}$. Subsequently, QwenVL is employed again to conceive suitable objects and corresponding editing instructions. These inputs are fed into the Qwen-Image-Edit model to synthesize the full composite image $I_{\text{img}}$. Finally, we reuse the extraction-removal operations from Pipeline 1 to derive the foreground layers.

**Occlusion Augmentation Pipeline (Pipeline 3).** To address the scarcity of occluded instances in existing data, we design a third pipeline focused on generating synthetic occlusion. In this setup, the full images $I_{\text{img}}$ produced by Pipeline 1 are utilized as the background layer, $I_{\text{bg}}$. Following a procedure similar to Pipeline 2, we employ QwenVL to conceive new object descriptions and editing instructions, followed by Qwen-Image-Edit to insert new objects into the scene. This process creates composite images with artificial occlusion relationships. To verify the presence of occlusions, we calculate the Intersection over Union (IoU) between all pairs of generated layers; a sample is retained if the IoU between any layer $I_{\text{fg}}^i$ and another layer $I_{\text{fg}}^j$ $(i \neq j)$ exceeds a threshold of 0.1.

**Quality Control and Dataset Splitting.** Prior to manual review, a rigorous automated filtering mechanism is applied to all pipelines. We compute the LPIPS distance between $I_{\text{img}}$ and $I_{\text{bg}}$ exclusively over the non-foreground regions, retaining only those samples where the LPIPS score is $\leq 0.1$. We opt for LPIPS over pixel-wise metrics like MSE because natural scene compositions often involve subtle effects such as shadows or reflections that bleed from the foreground into the background; while MSE is sensitive to these minor photometric shifts, LPIPS effectively evaluates perceptual fidelity.

For the evaluation set, **RevealLayerBench**, we conduct a rigorous secondary manual curation to select images characterized by rational layouts and harmonious background consistency. Furthermore, we manually balance the type distribution to incorporate a wide spectrum of challenging scenarios, including object occlusions, large-area targets, and transparent materials. To prevent data leakage, we strictly partition the dataset: since Pipelines 2 and 3 are synthesized based on the imagery from Pipeline 1, any image in the evaluation set necessitates the removal of its original source image from the training set. Ultimately, our benchmark comprises the **RevealLayer-100K** training set, consisting of 100K tuples $\{I_{\text{img}}, I_{\text{bg}}, \{I_{\text{fg}}^i\}_{i=1}^N, \{\text{Bbox}_i\}_{i=1}^N\}$, and the **RevealLayerBench** evaluation set of 200 high-quality images.

Figure 6 presents the statistical distributions of semantic categories and layer complexity within the RevealLayer-100k. As depicted in subfigure (a), the dataset exhibits a diverse and balanced composition across five major semantic categories. Subfigure (b) illustrates the layer complexity, characterized by a predominance of two-layer structures alongside a substantial inclusion of intricate multi-layer configurations. This distribution ensures that the training data encompasses both common compositional patterns and challenging structural complexities, thereby facilitating the development of robust layer decomposition models.

# B. Experimentation and Visual Analysis

In this section, we mainly show more experimental data and visual analysis of the role of the module.

*Table 7.* Quantitative comparison of object removal on OBER-Test. Comparative methods employ effect masks as guidance. For ObjectClear, reported metrics exclude the post-processing step of blending the background with the original image.

| Guidance | Method | OBER-Test | | | |
|---|---|---|---|---|---|
| | | PSNR↑ | SSIM↑ | LPIPS↓ | FID↓ |
| Effect-Mask | PowerPaint | 25.61 | 0.7359 | 0.0997 | 43.76 |
| | SmartEraser | 25.81 | 0.7361 | 0.0931 | 38.92 |
| | RORem | 26.88 | 0.7566 | 0.0978 | 39.47 |
| | AttentiveEraser | 28.27 | 0.8657 | 0.1166 | 35.54 |
| | ObjectClear* | 27.25 | 0.7645 | 0.0875 | 32.85 |
| Box | RevealLayer (Ours) | **30.16** | **0.9153** | **0.0694** | **25.62** |

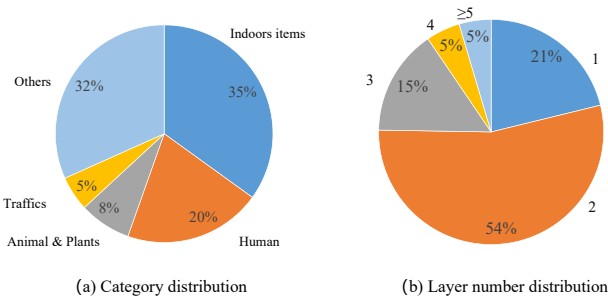

(a) Category distribution      (b) Layer number distribution

*Figure 6.* The category distribution and layer number distribution of our RevealLayer-100k.

## B.1. Object Removal

Conventional object removal models typically necessitate refined masks encompassing object effects (e.g., shadows and reflections) for optimal performance. We conducted supplementary experiments on the OBER-Test benchmark, providing baseline methods with effect masks as input, while our method utilizes only coarse bounding box guidance. As presented in Table 7, although the provision of effect masks offers baselines a distinct advantage, these methods continue to exhibit distorted residues and artifacts, failing to eradicate object-induced effects fully. Moreover, the expanded scope of the masks often leads to the inadvertent elimination of valid scene elements. Notably, for Object-Clear, strictly enforcing external effect masks compromises its intrinsic attention mechanism for mask prediction, resulting in a slight performance degradation. In stark contrast, our method achieves state-of-the-art performance relying solely on coarse bounding boxes, effectively eliminating object effects while reconstructing backgrounds with superior textural fidelity.

*Table 8.* Quantitative results on the PrismLayersPro validation set. RevealLayer is fine-tuned for 4k steps on PrismLayers.

| Method | PSNR↑ | SSIM↑ | FID↓ | IoU↑ | F1↑ |
|---|---|---|---|---|---|
| CLD | 27.646 | 0.874 | **19.413** | **0.867** | **0.920** |
| RevealLayer | **28.360** | **0.889** | 22.716 | 0.845 | 0.903 |

## B.2. Generalization to Stylized Images

To evaluate cross-domain generalization, we conduct supplementary experiments on the PrismLayersPro validation set of stylized poster images. After only 4k fine-tuning steps on PrismLayers, RevealLayer achieves higher PSNR and SSIM than CLD, indicating strong structural reconstruction under domain shift. CLD performs better on FID, IoU, and F1, likely due to its closer alignment with stylized appearance and alpha-layer distributions. These results show that RevealLayer remains competitive on non-photorealistic images with limited fine-tuning.

## B.3. Controllable Layer Decomposition

The RevealLayer we proposed framework utilizes user-specified bounding boxes to support decomposition with high degrees of freedom. As illustrated in Figure 7, instances specified by 1-3 input bounding boxes are decomposed into foreground layers, while all other objects are correctly classified as background. Notably, the first example in Row 1 and the second example in Row 3 demonstrate that the method accurately distinguishes foreground from background and reconstructs occluded regions with structural integrity in complex scenes, even when provided only with the bounding box of an occluded instance.

## B.4. Domain Adaptation for VAE Reconstruction

The original VAE employed in the ART architecture was primarily trained on graphic design and vector-style datasets. Consequently, a domain gap exists when applying this model to natural scenes, which are characterized by complex lighting and high-frequency textures. To mitigate this discrepancy and enhance reconstruction fidelity for our task, we finetuned the decoder of the pre-trained VAE on the RevealLayer-100k dataset, denoted as **XVAE**.

To assess the effectiveness of this adaptation, we measured the reconstruction quality on a multi-layer test set of natural scene images.. As presented in Table 9, the finetuned model demonstrates consistent improvements across all metrics compared to the original baseline. Specifically, we observe gains in both pixel-wise fidelity (PSNR, SSIM) and perceptual quality (LPIPS, FID) for full images, as well as separated background and foreground layers. These results confirm that bridging the domain gap in the autoencoder stage is crucial for preserving details in natural image generation.

## B.5. Additional analysis of RevealLayer

### B.5.1. ANALYSIS OF COMPLEX LAYER DECOMPOSITION

To evaluate the performance of layer separation, we conduct a comparative analysis between our method and CLD on the

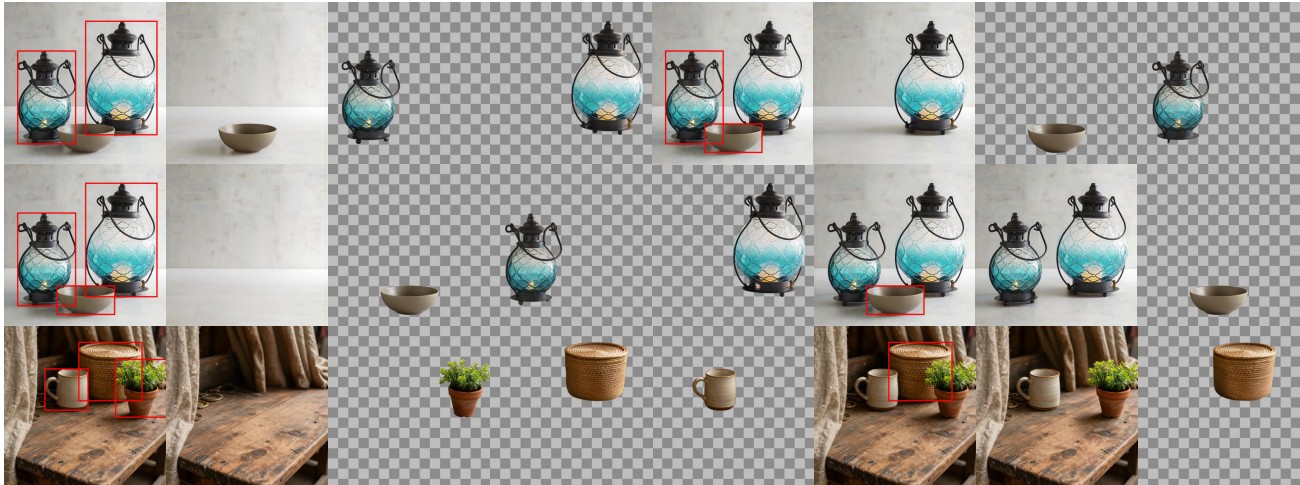

*Figure 7.* Qualitative results of controllable layer decomposition. Our method consistently decomposes the desired layers, regardless of the number or location of selected regions.

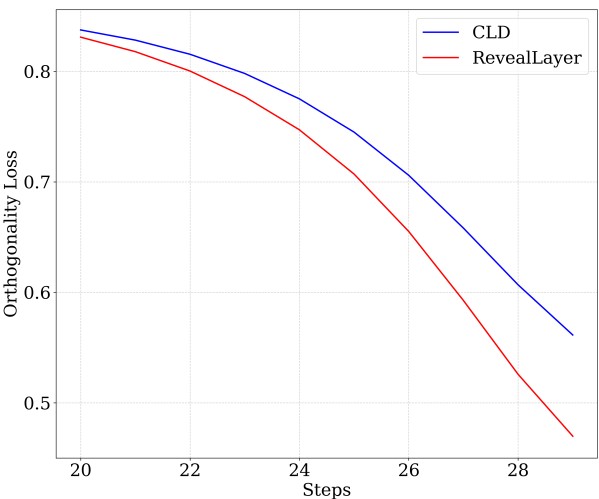

*Figure 8.* Comparison of the degree of layer separation. This shows the orthogonality loss values for each denoising step during inference on RevealBench.

*Table 9.* Quantitative comparison of VAE reconstruction quality before and after finetuning on natural scenes.

| Setting | PSNR↑ | SSIM↑ | LPIPS↓ | MSE↓ | FID↓ | rFID↓ |
|---------|-------|-------|--------|------|------|-------|
| *Full Image* | | | | | | |
| ART | 41.55 | 0.987 | 0.013 | 0.087 | **1.183** | **0.037** |
| XVAE | **41.74** | **0.987** | **0.011** | **0.086** | 1.227 | 0.038 |
| *Background* | | | | | | |
| ART | 46.68 | 0.993 | 0.009 | 0.025 | 0.726 | 0.025 |
| XVAE | **48.74** | **0.994** | **0.006** | **0.016** | **0.584** | **0.021** |
| *Foreground* | | | | | | |
| ART | 41.89 | 0.982 | 0.031 | **2.339** | 4.140 | 0.339 |
| XVAE | **41.94** | **0.982** | **0.031** | 2.340 | **4.081** | **0.332** |

RevealBench dataset, utilizing the disentanglement metric defined in Eq. (15). As illustrated in Figure 8, with more denoising steps, our method increasingly matches the ground truth in separating background and foreground layers, outperforming CLD. This quantitative advantage confirms that our approach effectively minimizes feature leakage, resulting in superior semantic independence and more precise layer decomposition.

### B.5.2. ANALYSIS OF SHARED NOISE STRATEGY IN LAYER DECOMPOSITION

We formulate the image layer decomposition task as a joint modeling of variable-length sequences for different fore-

ground and background layers. Using shared noise for all foreground layers enforces a shared stochastic origin in latent space, which aligns with the joint flow modeling assumption of RevealLayer and implicitly regularizes inter-layer consistency, resulting in more stable occlusion completion and fewer cross-layer artifacts.

As reported in Table 10, this strategy yields performance improvements on RevealBench, specifically increasing the foreground PSNR by 0.18 and decreasing the FID by 0.23. We hypothesize that this approach introduces a beneficial inductive bias, facilitating the separation of inherent data differences between background and foreground layers. The spectral analysis in Figure 9 offers a physical explanation for this phenomenon: background information is predominantly concentrated in low frequencies, whereas foregrounds exhibit a richer high-frequency spectrum. This noise initialization strategy effectively enhances the model's capacity for complex layer decomposition, a direction we plan to investigate further in future work.

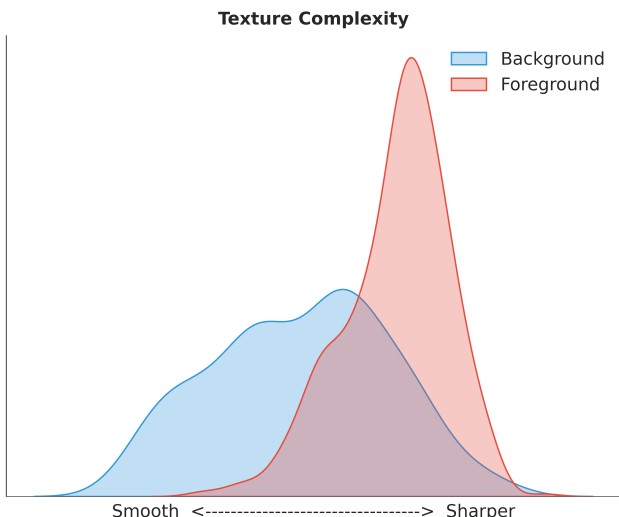

**Texture Complexity**

Smooth <--------------------------------> Sharper

*Figure 9.* Distribution of texture complexity for background and foreground regions. The x-axis represents the Log-Variance of the Laplacian operator, higher values indicate sharper images with more high-frequency edge information.

*Table 10.* Qualitative analysis on RevealLayerBench. Reveal-Layer* indicates that foreground layers share the same noise during inference.

| Method | Background-level | | | Foreground-level | | | |
|---|---|---|---|---|---|---|---|
| | PSNR↑ | LPIPS↓ | FID↓ | PSNR↑ | LPIPS↓ | FID↓ | SoftIoU↑ |
| RevealLayer | 25.53 | 0.1483 | 53.81 | 32.13 | 0.0217 | 18.42 | 0.9432 |
| RevealLayer* | **25.54** | **0.1479** | **53.48** | **32.31** | **0.0214** | **18.19** | **0.9433** |

## B.6. Q-Insight Metrics

To establish a quantitative method for evaluating the outputs of our approach and enabling objective comparison with other competing methods, we adopt the Q-Insight framework (Li et al., 2025b), the current state-of-the-art image assessment model, as used in CLD (Liu et al., 2025b). For a fair comparison, we follow CLD and evaluate the quality and editability of the generated layers along three primary dimensions. Specifically, we conduct a structured assessment of model outputs based on the following criteria:

- **Semantic Consistency:** Measures the semantic independence and completeness of each decomposed layer, as well as its alignment with the intended semantic content.

- **Visual Fidelity:** Evaluates visual quality, preservation of details (i.e., visual consistency with the original image), and overall realism.

- **Editability:** Assesses the manipulability of the decomposition to support subsequent editing, modification, and localized adjustments.

The evaluation prompt used is as follows:

---

**Q-Insight Prompt**

You are evaluating the quality of a layered image decomposition. The input consists of multiple images: [Original Image], [Background], [Layer 0], [Layer 1], ..., [Layer N].
**# — THINKING STEP —**
First, reason about the decomposition quality step-by-step inside the `<think>` tags. Analyze the semantic completeness of each layer, the visual fidelity of the layers and background, and the practical editability of the layer structure based on the criteria below.
**# — RATING STEP —**
Second, based on your reasoning, provide three separate ratings. All ratings should be floats between 1.00 and 5.00, rounded to two decimal places.
**# — CRITERIA —**
1. **semantic_consistency:** (1.00 = fragmented, meaningless layers; 5.00 = all layers are semantically whole and independent).
Each layer should represent a complete, logically independent object or part (e.g., a complete animal, a complete plate, a person, etc.), and the background should be clean and natural without any residues of layer objects.
Your rating needs to consider the degree of completeness of the objects in each layer image and the reasonableness of the background.
2. **visual_fidelity:** (1.00 = severe artifacts, missing pixels; 5.00 = the unoccluded parts of the layers and background in the original image are completely consistent with the original image).
*Note: You must ignore the inherent artistic style of the image (e.g., photo vs. cartoon).*
Your score should *only* reflect errors resulting from the decomposition process, such as halos, incorrect background inpainting, color bleeding, or missing pixels.
3. **editability:** (1.00 = useless, under- or over-decomposed; 5.00 = perfect granularity for editing).
*Preference: A finer-grained decomposition (more layers) is preferred and should receive a higher score, as long as the individual layers still represent complete semantic parts.*
Do not penalize for 'over-decomposition' if the resulting fine-grained layers are logical and useful for an editor (e.g., separating a title from a body text is better than keeping them as one layer).
**# — FORMATTING —**
Return the result in JSON format with the following

---

```
keys:
{
    "semantic_consistency": <score>,
    "visual_fidelity": <score>,
    "editability": <score>
}
```

### B.7. Manual Evaluation Criteria

We designed the evaluation criteria by reviewing existing literature and consulting both professional designers and a large pool of users. Specifically, the evaluation dimensions are divided into two main categories: Layer Numbers and background and foreground layer quality.

For a given layer decomposition model, we provide the required number of layers or specific regions, and assess the results as follows:

1. Layer Numbers: This dimension reflects the model's controllability, i.e., whether it can generate the specified number of layers. A score of 1 is assigned if the requirement is met, and 0 otherwise.

2. Background Quality: This dimension focuses on the completion of overlapping regions and the consistency of visible areas in the background. A fully satisfactory background receives 2 points, minor defects receive 1 point, and unsatisfactory results receive 0 points.

3. Foreground Quality: This dimension assesses the edge details of different foreground layers and the effectiveness of layer disentanglement. A fully satisfactory foreground receives 2 points, minor defects receive 1 point, and unsatisfactory results receive 0 points.

The evaluation team consists of professional evaluators with extensive domain knowledge and experience, enabling accurate and reliable assessments according to the defined criteria.

The final score is computed as:

$$S_{\text{LayerNums}} = w_0 \cdot \text{count}(0) + w_2 \cdot \text{count}(1),$$

$$S_{\text{Bg Q}}, \ S_{\text{Fg Q}} = \sum_{i=0}^{2} w_i \cdot \text{count}(i),$$

where the weights are set to $w_0 = 0$, $w_0 = 0.5$, $w_2 = 1.0$.

## C. More Visual Results

### C.1. Removal Results

Figures 10 and 11 visualize qualitative comparisons of object removal results produced by our RevealLayer on the OBER-Test and RevealLayerBench datasets. The OBER-Test primarily targets the removal of single small objects, whereas RevealLayerBench encompasses more challenging scenarios involving multiple objects, large-area occlusions, and complex illumination. As observed, PowerPaint, SmartEraser, RORem, and AttentiveEraser struggle to effectively accomplish object removal; they fail to eliminate even minor residual shadows and frequently introduce extraneous objects or visible artifacts within the inpainted regions. While ObjectClear demonstrates reasonable performance in most scenarios, it tends to generate redundant content when dealing with large-area occlusions and fails to reconstruct complex illumination effects faithfully. In contrast, our method not only thoroughly eradicates target objects along with their accompanying shadows and reflections but also exhibits superior performance in complex scenarios characterized by multiple objects or large-area occlusions, thereby demonstrating exceptional adaptability and robustness.

### C.2. Matting Results

Figures 12 and 13 illustrate qualitative comparisons between our approach and generic segmentation models (SAM-H, SAM2-L, SAM3) alongside the specialized matting model MAM, evaluated on the AIM500 and RefMatte-RW100 datasets. While generic segmentation models perform adequately in simple scenarios, they suffer from three critical limitations: reliance on post-processing to refine hard edges, inability to segment large-scale objects effectively, and failure to capture fine-grained details. Although MAM excels at handling intricate edges, it frequently suffers from visible artifacts. In contrast, our method consistently demonstrates superior performance across diverse challenging scenarios.

### C.3. Layer Decomposition Results

Figure 14-17 present additional qualitative results for the layer decomposition task. Figure 14 illustrates scenarios involving extensive reflections and occlusions. Figures 15 and 16 demonstrate the decomposition of densely populated and structurally intricate environments, where multiple overlapping objects are disentangled into distinct layers. Furthermore, Figure 17 showcases instances of region-specific manipulation, where targeted edits maintain consistency with the surrounding context. Collectively, these figures cover a diverse range of scene complexities, including outdoor reflections and indoor object arrangements, demonstrating RevealLayer's generalization ability and robustness to varying scene complexities.

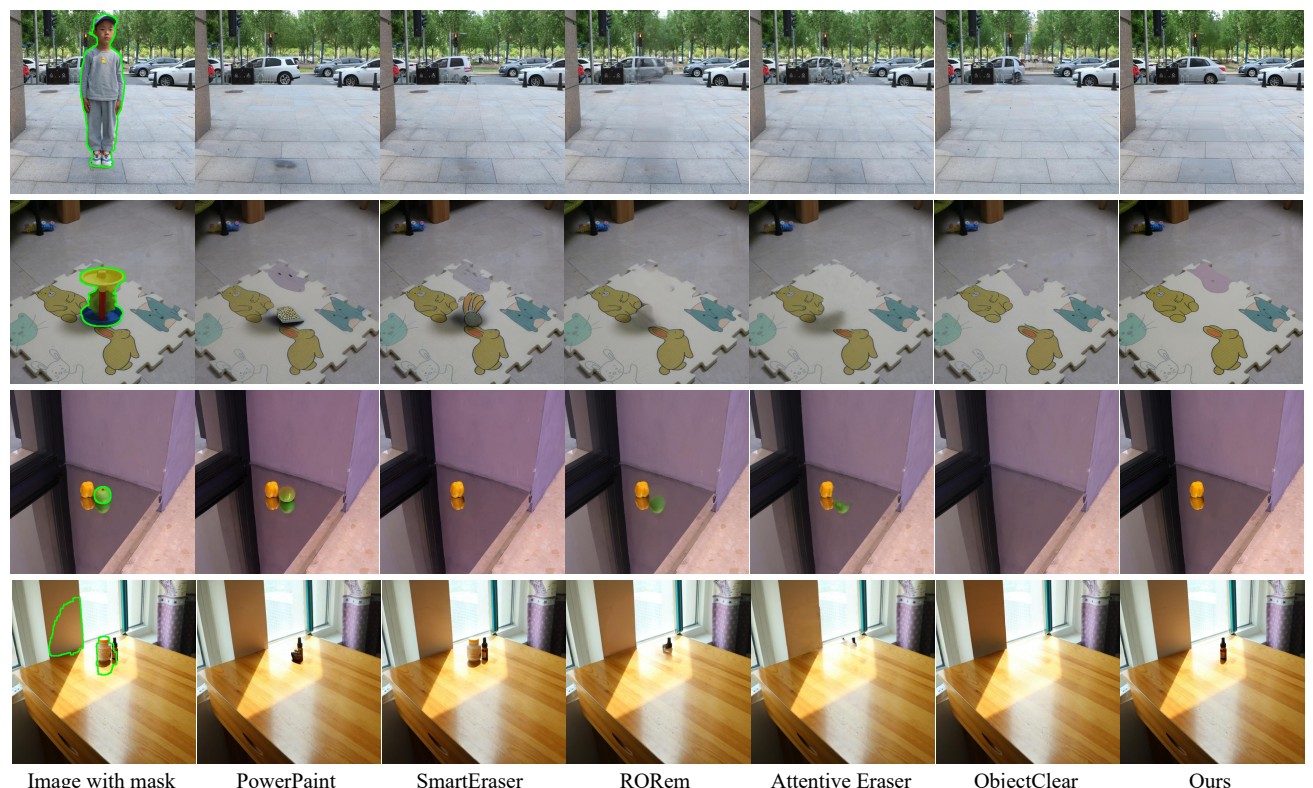

Image with mask    PowerPaint    SmartEraser    RORem    Attentive Eraser    ObjectClear    Ours

*Figure 10.* Visual results in OBER-Test.

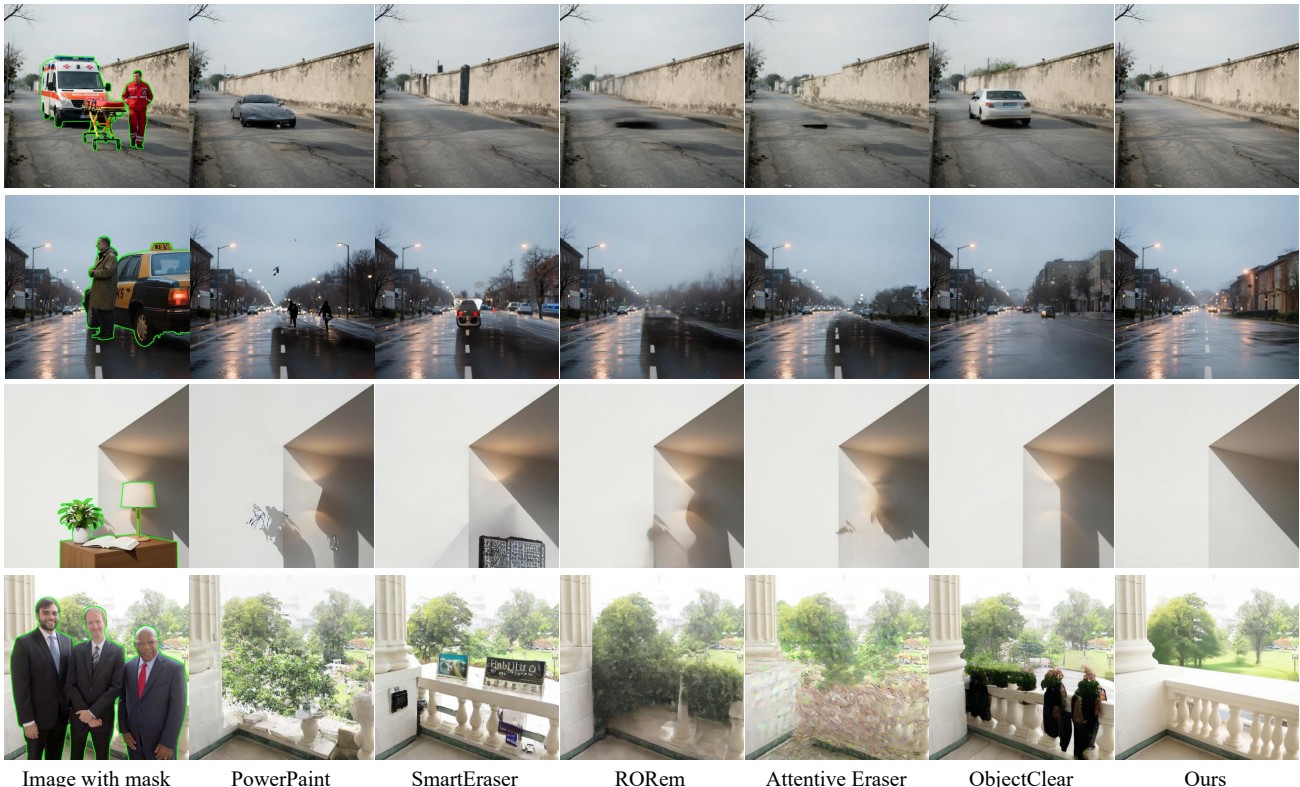

Image with mask    PowerPaint    SmartEraser    RORem    Attentive Eraser    ObjectClear    Ours

*Figure 11.* Visual results in RevealLayerBench.

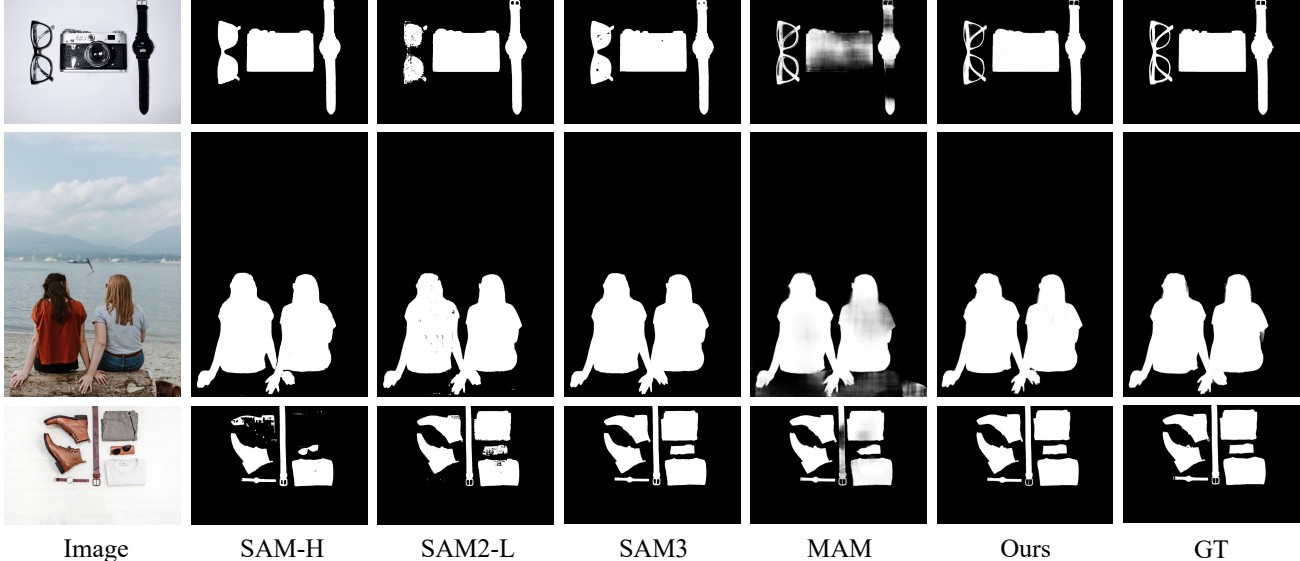

| Image | SAM-H | SAM2-L | SAM3 | MAM | Ours | GT |

*Figure 12.* Visual results in AIM500.

| Image | SAM-H | SAM2-L | SAM3 | MAM | Ours | GT |

*Figure 13.* Visual results in RefMatte-RW100.

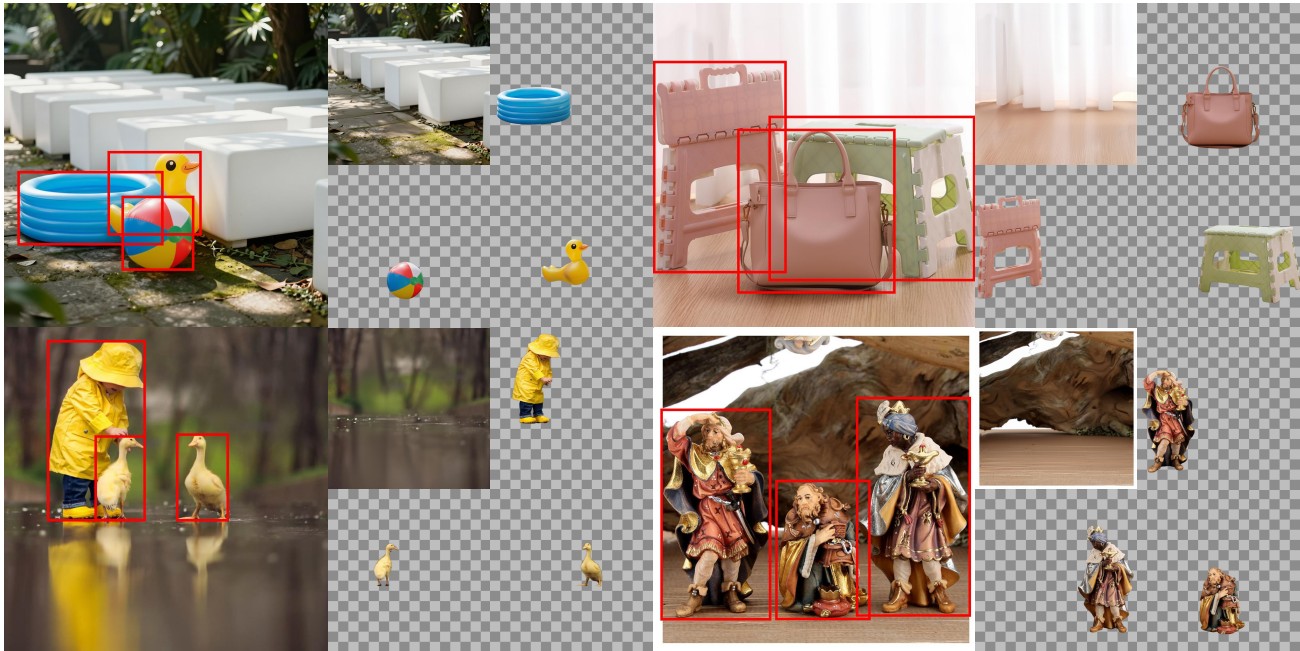

*Figure 14.* Additional visual results of layer decomposition.

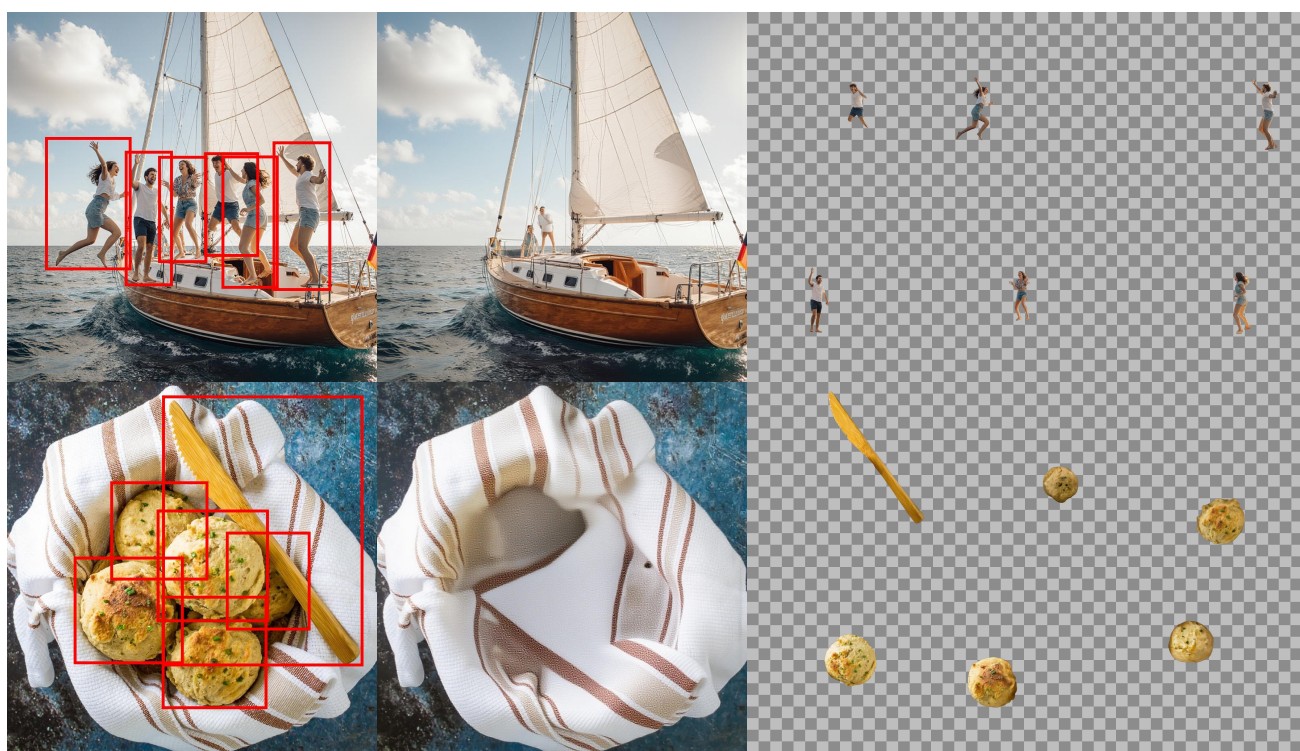

*Figure 15.* Additional visual results of layer decomposition.

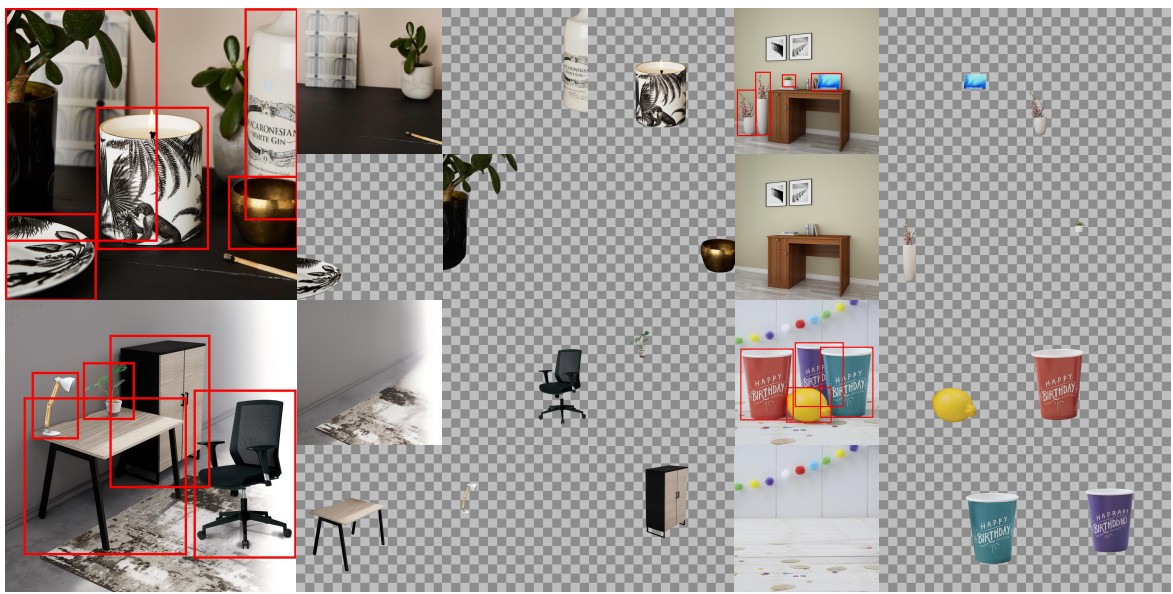

*Figure 16.* Additional visual results of layer decomposition.

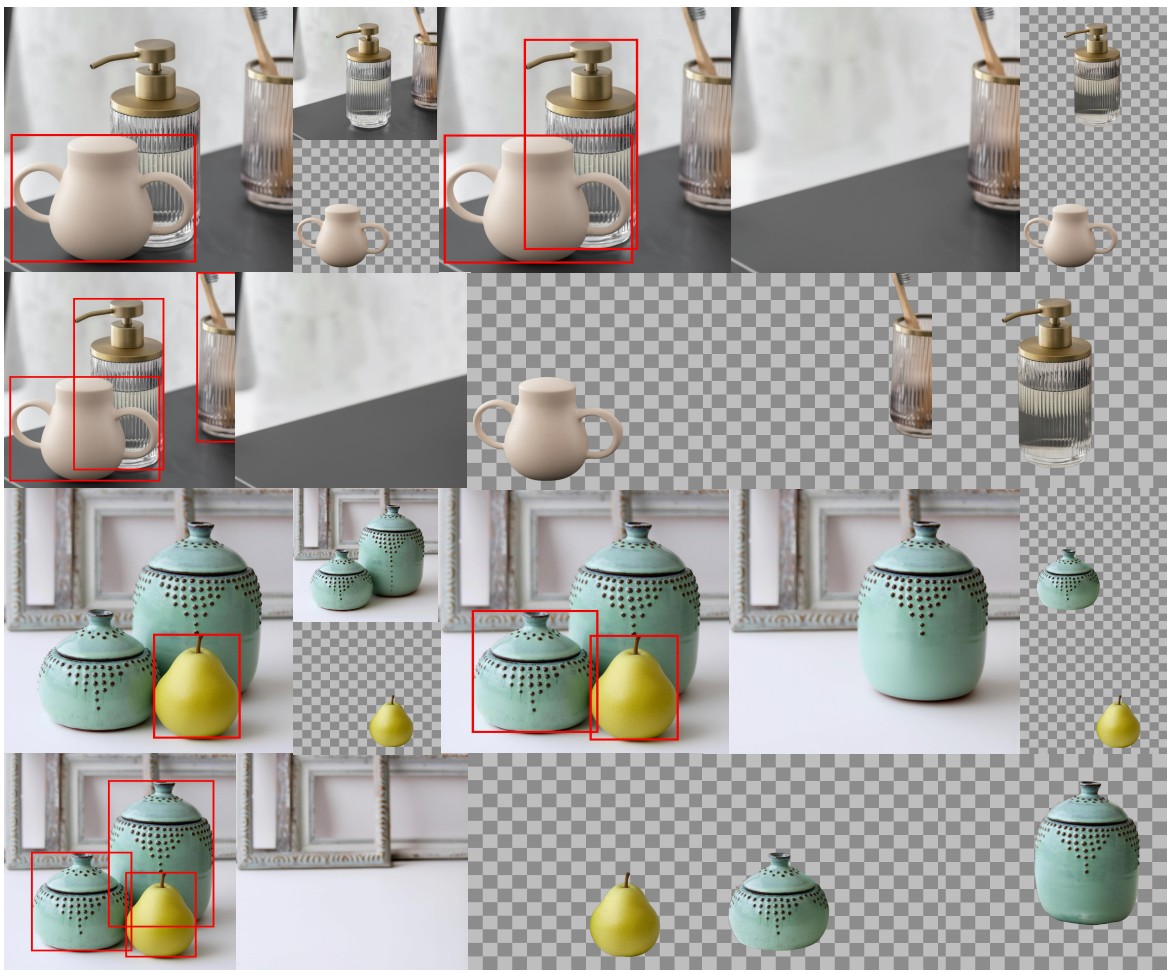

*Figure 17.* Additional visual results demonstrating the controllability of layer decomposition.

