# OpenReview forum: "RevealLayer: Disentangling Hidden and Visible Layers via Occlusion-Aware Image Decomposition"
_ICML.cc/2026/Conference — ICML 2026 regular_

### Official Review · Reviewer_yzzp · 2026-03-09

**Soundness:** 4
**Presentation:** 2
**Significance:** 3
**Originality:** 2
**Overall Recommendation:** 4
**Confidence:** 3

**Summary:**

This paper studies controllable layered image decomposition for natural images: given an RGB image and user-provided bounding boxes, the model predicts one background layer and multiple RGBA foreground layers, including occluded content. The method is built on a FLUX/MM-DiT backbone with a fine-tuned transparent-image VAE, and treats decomposition as variable-length latent sequence modeling over the background and cropped foreground regions. The main technical additions are Region-Aware Attention (RAA), which restricts cross-layer token interactions, an Occlusion-Guided Adapter (OGA), which injects masked image context to improve overlapping-region completion, and two auxiliary losses for alpha sharpness and background/foreground decoupling. The paper also introduces RevealLayer-100K and RevealLayerBench, and evaluates on object removal, matting, and multi-layer decomposition, where the proposed method reports clear gains over recent baselines.

**Compliance With Llm Reviewing Policy:**

Affirmed.

**Final Justification:**

My issue has largely been resolved. However, taking the comments from other reviewers into account, I believe the current score is fair, so I decide to maintain the score.

**Key Questions For Authors:**

- How robust is the proposed method to imperfect user-provided bounding boxes, such as shifted boxes, overly loose boxes, overly tight boxes, or boxes that cover multiple nearby instances? Could the authors provide a controlled robustness study under such perturbations?
- Can the authors provide a more detailed analysis of the data composition and potential bias in RevealLayer-100K / RevealLayerBench？
- Could the authors provide a more systematic analysis of failure cases, including representative examples and categorization of common failure modes？

**Limitations:**

yes

**Strengths And Weaknesses:**

Strengths:

- The paper studies controllable layered decomposition in natural images, requiring recovery of background, foreground RGBA layers, and occluded content. This setting is more aligned with practical editing needs than standard inpainting or matting, making the problem itself meaningful.
- The evaluation is relatively broad: in addition to multi-layer decomposition, the paper also assesses object removal and foreground matting, which helps validate the method from multiple related perspectives rather than only a single benchmark.
- The ablation study suggests that RAA, OGA, OrthLoss, and AlphaLoss all contribute positively to final performance, indicating that the gains are not solely due to a strong backbone but also to the proposed design choices.


Weaknessnes:
- The paper assumes reasonably accurate user boxes, but does not analyze robustness to shifted, loose, overly tight, or ambiguous boxes that cover multiple instances. This is an important missing evaluation for real interactive use.
- The realism of RevealLayer-100K may be influenced by the generative and automated data pipeline used to construct it. Since the dataset depends on multiple external models for background editing, segmentation, matting, and filtering, the resulting distribution may inherit model-specific artifacts, stylistic biases, or restoration traces. The paper does not sufficiently analyze whether such biases make the benchmark especially favorable to the proposed method rather than representative of truly independent real-world data.
- The paper predominantly emphasizes successful examples, but it provides limited systematic characterization of failure cases. A clearer discussion of when and why the method fails would help readers better understand its operating regime and limitations.

---

> ### Author Rebuttal · Authors · 2026-03-30
>
> We really appreciate your detailed review and valuable suggestions. We address each of your comments below.
>
> W1&Q1: Bounding Box Robustness
>
> A1: We conducted experimental verification on different types of input boxes. This issue overlaps with reviewers Wt8J-W1 and rbJc-L1.
> 1. As shown in Reb-Table 2 in Review rbJc-L1. (a) Excessive boxes (≤20%), small offset (≤5%) and  slightly inadequate (≤5%) lead to negligible degradation.  (b) Larger offset (5–10%) and significantly inadequate (5–10%)  cause noticeable drops due to misalignment.
> 2. Multi-instance coverage. In the main paper (Fig. 5) and Appendix (Figs. 14–17),  we shows that partial coverage of multiple instances is well handled, while full coverage may lead to some degree of performance degradation.
> 3. In real applications, we leverage GroundingDINO, SAM, or MLLMs for automated instance detection, converting flexible inputs (text/point/mask/bbox) into accurate bounding boxes. This decouples detection from layer decomposition, improving robustness and flexible controllability.
>
> W2&Q2:  Dataset Bias Concern
>
> A2:   We appreciate the reviewer's concern regarding data quality and model robustness.
> 1. Dataset quality and diversity: While our dataset is constructed with automated components, we mitigate potential biases through a dual-stage validation process, including a rigorous automated filtering pipeline and manual verification by expert annotators focusing on boundaries, occlusion completion, and visual consistency. Appendix A further demonstrates the diversity of data sources and three distinct data construction pipelines.
> 2. Cross-dataset validation: Our method demonstrates strong performance on independent real-world datasets (OBER-Test, AIM-500, RefMatte-RW100) and  performs strongly on RevealLayerBench, showing it does not rely on the pipeline-specific distribution.
> 4. Failure analysis. Based on human evaluation (Table 5) and offline inspection of failure cases (will be included in the revision), failures do not appear concentrated on specific artifact patterns, indicating robustness beyond pipeline bias.
>
> We will include additional analysis of potential biases in the revision.
>
> W3&Q3:  Failure Cases Analysis
>
> A3:  This is an excellent suggestion, a comprehensive analysis must include failure cases. We conduct experiments and analyses on abnormal input boxes and complex scenarios. This issue overlaps with reviewers Wt8J-W3 and rbJc-L1.
> 1. Inaccurate or ambiguous boxes. We evaluate robustness to abnormal user inputs (e.g., shifted or inaccurate bounding boxes) in Reb-Table 2 (review rbJc-L1). Results e4 (5%–10% offset) and e6 (5%–10% inadequate) show that severe misalignment degrades performance, as our method crops the bbox to reduce token length for efficiency. This can be mitigated by a preprocessing step using GroundingDINO or SAM to refine user inputs.
> 2. Severe occlusion. A common failure occurs when the target is heavily occluded, especially in cross-occlusion or when the occluder covers a large area. In these cases, very little of the target is visible, making it difficult to determine complete the occluded content.
> 3. Complex textured backgrounds. Failures may occur on dense or repetitive high-frequency textures (e.g., grass, foliage, clutter). These regions are inherently ambiguous, with many plausible ways to reconstruct details, and are underrepresented in training data, making accurate recovery more challenging.
> 4. Complex transparent or translucent regions. Transparent materials such as frosted glass are particularly challenging because foreground and background appearances are intrinsically mixed, and the alpha boundary is often soft or ambiguous. This may result in incomplete separation or inaccurate alpha estimation.
> 5. Despite these challenges, the model performs well on some transparent cases (Figs. 5, 7), text recovery (Fig. 1), and small objects (Fig. 2), demonstrating robustness in certain difficult scenarios.
>
> We will expand the revision to systematically characterize failure modes, better clarifying the operating regime and limitations of our method.

---

> > ### Author Rebuttal · Reviewer_yzzp · 2026-04-04
> >
> > My issue has largely been resolved. However, taking the comments from other reviewers into account, I believe the current score is fair, so I decide to maintain the score.

---

> > > ### Author Response · Authors · 2026-04-07
> > >
> > > Thank you for the update and for recognizing the value of our work. We appreciate your continued positive assessment, and we are glad that our rebuttal helped address your concerns.

---

### Official Review · Reviewer_44mu · 2026-03-10

**Soundness:** 2
**Presentation:** 2
**Significance:** 2
**Originality:** 2
**Overall Recommendation:** 3
**Confidence:** 3

**Summary:**

This paper proposes RevealLayer for image layer decomposition. RevealLayer consists of three key technical designs: (1) Region-Aware Attention that imposes structural constraints on token interactions to disentangle visible and hidden layer content and reduce inter-layer feature interference; (2) Occlusion-Guided Adapter that leverages localized contextual information from the original image to enhance semantic coherence in overlapping occluded regions; and (3) Hard-constraint alpha loss and soft-constraint orthogonality loss to enforce sharp alpha boundaries and suppress residual artifacts between layers. To support model training and systematic evaluation, the authors also develop a rigorous multi-pipeline data curation workflow and construct RevealLayer-100K, a large-scale multi-layer natural image dataset, along with RevealLayerBench, a dedicated benchmark for layer decomposition in complex natural scenes. Extensive experiments demonstrate the superiority of RevealLayer against existing methods.

**Compliance With Llm Reviewing Policy:**

Affirmed.

**Key Questions For Authors:**

- Will the authors publicly release the RevealLayer-100K dataset, RevealLayerBench benchmark, and the code for the data construction pipeline?
- CLD is trained for stylized poster images, and it is unfair to compare it on RevealLayerBench directly. How about using RevealLayer for the stylized poster images? This is conducive to understanding its generalization ability.
- The paper reports the overall inference time of RevealLayer, which seems to take a long time. How about the computational overhead introduced by RAA and OGA?

**Limitations:**

No limitation. Discussing the limitations of the proposed method will help to provide a comprehensive understanding of it.

**Strengths And Weaknesses:**

Strengths
- The proposed method points out the issues of existing layer decomposition methods with several technical designs.
- The authors introduce RevealLayer-100K and RevealLayerBench, addressing the lack of large-scale datasets for complex natural scenes.
- Extensive experiments demonstrate that RevealLayer consistently outperforms state-of-the-art methods.

Weaknesses
- The author points out that region-level layer disentanglement and intermediate feature enhancement play a crucial role in improving layer decomposition and occlusion completion. Is there any evidence to support this phenomenon?
- This method is a bbox-guided approach, while the other methods are mask-guided approaches.  How will these two types of guidance affect performance? Can this method be modified to be mask-guided, thereby providing relevant evidence to prove that the gains achieved are not due to the change in the guidance?
- Insufficient experimental details, e.g., how do the authors select the weights of the hard-constraint alpha loss and soft-constraint orthogonality loss?

---

> ### Author Rebuttal · Authors · 2026-03-30
>
> We sincerely thank the reviewer for your constructive comments. We address your concerns below.
>
> W1: Effectiveness of RAA and OGA
>
> A1:  Table 4 of the paper provides a systematic ablation study. Starting from the same flux-only baseline, adding RAA significantly improves background PSNR and foreground SoftIoU, indicating enhanced region-level disentanglement. OGA improves PSNR and FID for both background and foreground layers, demonstrating stronger occlusion completion capability.
> Detailed results (e1, e2, e5, e6) in Reb-Table 2 (review rbJc-L1) show that Flux-inpaint or Flux-only baselines are limited for the complex multi-layer decomposition task. We also visualize intermediate features and observe that RAA and OGA lead to clearer layer and more semantically consistent occlusion completion. These visualizations will be included in the revised version.
>
> W2: BBox vs. Mask Guidance Impact
>
> A2:  BBoxes provide a looser prior that better supports foreground preservation and occlusion completion, while mask guidance favors background recovery but may hinder foreground separation and completion. Our gains are not due to the choice of guidance.
>
> We conduct two experiments: (a) replace bbox with mask guidance at inference (with/without dilation); (b) fine-tune mask-based variants.
> From Reb-Table 3, bbox-guided baseline(e1) achieves the best performance. Mask guidance (e2) degrades foreground quality with jagged edges, while dilation mask (e3) partially recovers performance. Fine-tuned mask-guided variants (e4, e5) still underperform bbox guidance.
>
> Occlusions in natural scenes lack clear boundaries, making mask guidance unreliable. In contrast, bbox guidance is more robust, and avoids extra preprocessing and computational overhead without sacrificing performance.
>
> Reb-Table 3.  Quantitative results  on Bbox&Mask guidance.
> | Exp | Type | Setting              | BG         | BG         | BG       | FG         | FG         | FG       | FG         |
> |-----|------|----------------------|------------|------------|----------|------------|------------|----------|------------|
> |     |      |                      | PSNR ↑     | LPIPS ↓    | FID ↓    | PSNR ↑     | LPIPS ↓    | FID ↓    | SoftIoU ↑  |
> | e1  | bbox| base            | 22.08      | 0.155      | 59.54    | 30.62      | 0.025      | 20.86    | 0.9224     |
> | e2  | mask | base_mask            | 22.17      | 0.1517     | 56.15    | 28.87      | 0.033      | 34.78    | 0.888      |
> | e3  | mask | base_mask_dilated    | 22.17      | 0.1520     | 57.13    | 30.69      | 0.025      | 21.72    | 0.920      |
> | e4  | mask| base_FT_mask         | 23.68      | 0.1536     | 60.27    | 29.81      | 0.0287     | 33.06    | 0.9116     |
> | e5  | mask | base_FT_mask_dilated | 22.96      | 0.1714     | 64.50    | 28.93      | 0.0327     | 28.32    | 0.915      |
>
> W3: Experimental Details
>
> A3: We will include more detailed experimental settings in the revision. For loss weighting, flow matching loss is in 0.15–0.25, the orthogonality loss is below 0.01 (up to 0.1–0.2 under stronger penalties), and the alpha loss is ~0.1. Final weights are determined empirically to ensure they do not conflict.
>
> Q1: Data Open Source
>
> A4:  We will release RevealLayer-100K, RevealLayerBench, and the data construction pipeline. To our knowledge, this is the first large-scale, high-quality, high-resolution dataset for multi-layer decomposition in natural scenes.
>
> Q2: Model Generalization Capability
>
> A5:  By fine-tuning 4k steps on PrismLayers, RevealLayer achieves strong cross-domain performance on stylized poster data (see Reb-Table 4). As it prioritizes structural consistency(e.g., layer decomposition and occlusion reasoning) over style fitting, its alignment with domain-specific appearance is slightly weaker than CLD, resulting in a small FID gap.
>
> Reb-Table 4.  Quantitative results on PrismLayersPro Validation.
> | Method       | PSNR ↑ | SSIM ↑ | FID ↓  | IoU ↑ | F1 ↑  |
> |--------------|--------|--------|--------|-------|-------|
> | CLD          | 27.646 | 0.874  | 19.413 | 0.867 | 0.920 |
> | RevealLayer  | 28.360 | 0.889  | 22.716 | 0.845 | 0.903 |
>
> Q3: Runtime Analysis
>
> A6:   We evaluate the average runtime under the same setting (80G A100, FP16, 30 steps), as shown in Reb-Table 5. The Flux-Only baseline takes 93s (76%) for multi-layer generation, while the RAA attention mask adds 26s (21%). Enabling FlashAttention reduces latency by 19% to 99s, and further applying quantization and cache optimization reduces it by 53% to 57s with minimal performance loss.
>
> Reb-Table 5.  Runtime analysis of each component.
> | Flux-Only | RAA | OGA | RevealLayer-Full | RevealLayer-Full (FlashAttention) | RevealLayer-Full (FlashAttention, Int4, Cache) |
> |-----------|-----|-----|------------------|------------------------------------|-----------------------------------------------|
> | 93s       | 26s | 3s  | 122s             | 99s                                | 57s                                           |

---

> > ### Author Rebuttal · Reviewer_44mu · 2026-04-03
> >
> > Thanks for the rebuttal. But I find it difficult to agree with the importance of this work, so I decide to maintain the score.

---

> > > ### Author Response · Authors · 2026-04-04
> > >
> > > We thank the reviewer for the positive feedback and for acknowledging that our previous rebuttal has addressed the main concerns. We would also like to clarify the broader significance of this work.
> > >
> > > The significance of our work lies in addressing a key capability that is still missing in the current works. Existing natural-image layer decomposition methods mainly focus on either single-target layer extraction under referring conditions or foreground/background decomposition, such as RLD (ICLR 2026), From Inpainting to Layer Decomposition (CVPR 2026), and LayerDecomp (CVPR 2025). Qwen-Image-Layered achieves decomposition into multiple RGBA layers, but it is not designed for controllable decomposition; moreover, similar to CLD, its data setting is closer to layered PSD files and design-oriented imagery than to our target problem of controllable decomposition in unconstrained natural scenes. Compared with stylized imagery, natural scenes involve more complex lighting effects, textures, and entangled inter-layer interactions. In this sense, our contribution is not merely another empirical improvement over existing methods, but an important step toward controllable multi-layer decomposition in general natural scenes within a unified framework.
> > >
> > > This problem is also practically important. Decomposing an image into a clean background layer and editable RGBA foreground layers provides a much more useful representation for downstream image editing than masks or a single edited RGB output. Such a layer-based representation enables more flexible and consistent object-level editing, which is exactly the kind of capability needed in real image manipulation workflows. We will revise the paper to make this practical significance clearer.
> > > In addition, we will release both our model and dataset, which we hope will facilitate future research on controllable layered decomposition.
> > >
> > > We are also encouraged that the practical value and significance of this research direction have been recognized by other reviewers. They shared the view that our setting aligns well with actual editing needs, and agreed that this is a challenging and relatively underexplored task with meaningful applications. Furthermore, it was acknowledged that lowering the user's operational threshold and introducing the proposed dataset are of great significance to the advancement of the field. We hope this consensus on the value of the problem itself helps further illustrate the underlying meaningfulness of our work.

---

### Official Review · Reviewer_rbJc · 2026-03-12

**Soundness:** 3
**Presentation:** 3
**Significance:** 3
**Originality:** 3
**Overall Recommendation:** 5
**Confidence:** 3

**Summary:**

This paper proposes RevealLayer, a diffusion-based framework for decomposing a single RGB image into multiple RGBA layers with explicit transparency and occlusion completion. The model take an image and user-specified bounding box and predicts a background layer and multiple foreground layers. The method introduce three main components: Region-Aware Attention (RAA), Occlusion-Guided Adapter (OGA), and composite loss functions. The paper also introduce a new datasets (RevealLayer-100K) and evaluation benchmark (RevealLayerBench) for multi-layer natural image decomposition.

**Compliance With Llm Reviewing Policy:**

Affirmed.

**Final Justification:**

Thank you to the authors for the rebuttal. My concerns have been addressed, so I have raised my rating.

**Key Questions For Authors:**

If the concerns are well addressed, I would be willing to further raise the rating to 5 or 6.

**Limitations:**

Discussing the limitations or failure cases would further strengthen the work.

**Strengths And Weaknesses:**

### Strengths
1. Layered decomposition of natural images with occlusion completion is a challenging and relatively underexplored task with application in image editing and scene manipulation.
2. The proposed components (RAA and OGA) are well-motivated and directly target common issue such as cross-layer feature leakage and inconsistent reconstruction in occluded region.

### Weakness
1. The method requires user-provided bounding boxes, limiting full automation and potentially reducing applicability in complex scene. Is it possible to explore automatic instance detection instead of relying on user-specified box?
2. The dataset is largely constructed using automated pipelines and synthetic editing, which may introduce artifacts or biases. Could author provide more details about the dataset quality and annotation validation?
2. Could author consider briefly discussing image composition literature in Related Work, as it is closely related to layered RGBA representations. A short subsection could help better position the method.
4. The related work could be further strengthened by briefly discussing prior work on image composition and object insertion [1-5]. They are closely related to compositional image editing, and including them would help better position the proposed approach within the broader literature.

[1] Paint by Example: Exemplar-based Image Editing with Diffusion Models

[2] TF-ICON: Diffusion-based Training-free Cross-domain Image Composition

[3] AnyDoor: Zero-shot Object-level Image Customization

[4] Insert Anything: Image Insertion via In-Context Editing in DiT

[5] Does FLUX Already Know How to Perform Physically Plausible Image Composition?

---

> ### Author Rebuttal · Authors · 2026-03-30
>
> We thank you for your affirmation and constructive comments.
>
> W1: Input Automation and Scalability
>
> A1: In practice, we leverage GroundingDINO, SAM, or MLLMs for automated instance detection, enabling flexible inputs (text/point/mask/bbox) to be converted into accurate bounding boxes for our framework.  Using boxes is a practical and efficient design choice, offering stronger controllability than fully automated approaches (e.g., QwenImageLayer) while being cheaper to obtain than masks.
> Removing box priors entirely leads to a different task setting, requiring additional considerations in computational cost and model design. Fully automatic decomposition is an important direction as a  potential future work.
>
> W2: Quality and Annotation Details of RevealLayer-100K
>
> A2: We perform dual-stage dataset validation, and the model demonstrates stable performance on independent benchmark datasets.
> 1. Appendix A provides detailed data construction procedures. The dataset undergoes multi-layer image cleaning with a rigorous automated filtering mechanism, followed by manual verification by a team of five experts. Human annotation focuses on foreground boundary accuracy, object completeness, and the quality of background occlusion completion.
> 2. Our model consistently achieves strong performance on independent real-world datasets (OBER-Test, AIM-500, RefMatte-RW100) across different tasks, demonstrating robustness beyond our dataset.
> 3. We will release the dataset to support community research and validation.
>
> W3&W4: related works
>
> A3: We thank the reviewer for the suggestion. Image composition and object insertion are indeed related through layered RGBA representations. While these works mainly focus on foreground-background synthesis, our task emphasizes multi-layer decomposition and occlusion reasoning from a single image. We agree that including them would help better position our work, and will add a brief discussion in the revision.
>
> L1:  Failure Cases Analysis
>
> A4:  We observe failure modes including misaligned input boxes, and complex occlusion scenes.  This issue overlaps with reviewers  Wt8J-W3 and yzzp-W3.
> 1. Inaccurate boxes. Detailed analysis of box abnormalities is provided in Reb-Table 2. Mild perturbations such as e2 (excessive boxes within 20%), e3 (≤5% offset), and e5 (≤5% inadequate), lead to negligible performance degradation. In contrast, more severe perturbations, such as e4 (5%–10% offset) and e6 (5%–10% inadequate), result in noticeable drops.
> 2. Severe occlusion. Failures may occur when the target is largely hidden, particularly under cross-occlusion or when the occluder covers a large area. With limited visible evidence, layer ordering and occlusion completion become more ambiguous.
> 3. Complex transparent or translucent regions. Failures may occur in scenes with materials such as frosted glass, where foreground and background are inherently mixed and boundaries are not clearly defined. This can lead to incomplete separation or less accurate alpha estimation.
> 4. Complex texture completion. Failures may occur when recovering dense or repetitive high-frequency textures (e.g., grass, foliage, clutter). These regions lack clear structural cues and admit multiple plausible reconstructions, often leading to inconsistency or artifacts. In addition, such patterns are underrepresented in training data, further increasing the difficulty of accurate completion.
> 5. Despite these challenges, the model still succeeds in several challenging cases, including transparent scenes (Figs. 5, 7), text recovery (Fig. 1), and small objects (Fig. 2).
>
> In the revision, we will provide a more systematic analysis, including categorization, visualization, and quantitative evaluation, to better characterize the applicability and limitations of our method.
>
> Reb-Table 2.  Quantitative results of abnormal input box types on model performance.
> | Exp | Type              | BG         | BG         | BG       | FG         | FG         | FG       | FG         |
> |-----|-------------------|------------|------------|----------|------------|------------|----------|------------|
> |     |                   | PSNR↑      | LPIPS↓     | FID↓     | PSNR↑      | LPIPS↓     | FID↓     | SoftIoU↑   |
> | e1  | precise bbox      | 25.53      | 0.1483     | 53.81    | 32.13      | 0.0217     | 18.42    | 0.9432     |
> | e2  | excessive 10%-20% | 25.30      | 0.1507     | 54.28    | 31.69      | 0.0253     | 18.77    | 0.9368     |
> | e3  | offset 0-5%       | 25.30      | 0.1518     | 56.31    | 30.92      | 0.0260     | 20.22    | 0.9278     |
> | e4  | offset 5-10%      | 24.57      | 0.1615     | 61.36    | 27.22      | 0.0425     | 28.40    | 0.8569     |
> | e5  | inadequate 0-5%   | 25.26      | 0.1515     | 56.06    | 31.55      | 0.0237     | 19.22    | 0.9373     |
> | e6  | inadequate 5-10%  | 24.64      | 0.1610     | 62.36    | 28.01      | 0.0396     | 28.96    | 0.8757     |

---

> > ### Author Rebuttal · Reviewer_rbJc · 2026-04-04
> >
> > Thank you to the authors for the rebuttal. My concerns have been addressed, so I have raised my rating.

---

> > > ### Author Response · Authors · 2026-04-07
> > >
> > > Thanks for your acknowledgment of our work. We sincerely appreciate your decision to raise the score after reading our rebuttal, and we are pleased that our response helped clarify the paper.

---

### Official Review · Reviewer_Wt8J · 2026-03-12

**Soundness:** 2
**Presentation:** 3
**Significance:** 2
**Originality:** 3
**Overall Recommendation:** 3
**Confidence:** 3

**Summary:**

This paper addresses the issues of existing methods in natural image layer decomposition, including difficulties in occluded content completion, poor robustness of inter-layer decoupling, insufficient foreground boundary accuracy, and the scarcity of high-quality multi-layer natural image datasets. It proposes a diffusion-based RevealLayer controllable image layer decomposition framework. Guided by user-specified bounding boxes, this framework decomposes a single RGB image into multiple RGBA layers. Its core components include a Region Aware Attention (RAA) module, an Occlusion Guided Adapter (OGA), and a composite loss function. It also constructs a large-scale RevealLayer-100K multi-layer natural image dataset containing 100,000 samples and a RevealLayerBench benchmark set. Extensive experiments demonstrate that this method comprehensively surpasses existing state-of-the-art methods in all quantitative metrics and visualization effects for tasks such as image layer decomposition, object removal, and image matting, achieving more accurate inter-layer decoupling, occluded region recovery, and boundary fidelity.

**Compliance With Llm Reviewing Policy:**

Affirmed.

**Final Justification:**

I still have some questions. So at this time, I decide to retain my initial rating.

**Key Questions For Authors:**

See weaknesses.

**Limitations:**

1. The proposed model still suffers from high inference costs (122 seconds of inference time and 60GB of VRAM usage for a single 1024-resolution image), limiting its performance to high-end professional hardware and making it unsuitable for consumer devices and real-time editing scenarios.
2. The paper completely lacks discussion on the conditions under which the model fails. The authors should provide visual examples and failure case analyses, such as performance when handling highly complex transparent materials (e.g., overlapping frosted glass), dense text, or highly blurred occlusion boundaries.

**Strengths And Weaknesses:**

Strengths:
1. This paper defines the layer decomposition task as a bbox-guided controllable RGBA layer decomposition. Compared with existing methods that rely on fine masks, this reduces the user's operational threshold and solves problems such as error accumulation in cascaded methods and insufficient controllability in end-to-end methods.
2. By introducing RAA and OGA, "feature contamination" between layers is eliminated at the root of the model architecture, and the loss function precisely targets alpha edges and residual artifacts.
3. This paper proposes a new high-quality dataset of around 100K, which is of great significance to the advancement of the entire field.
4. The paper is clearly written and structurally complete.

Weaknesses:
1. All experiments in this paper use accurate bboxes from the dataset as guides. However, in real-world applications, user-provided bboxes often suffer from offsets, excessive/inadequate sizes, or inaccuracies. The paper does not conduct ablation experiments to assess the noise robustness of the bboxes, thus failing to verify the stability of the method in real-world scenarios.
2. The backbone of the model is FLUX.1 [dev], which has 12 billion parameters. However, in tasks such as object removal, the authors conduct a comprehensive comparison with baseline models like PowerPaint, SmartEraser and ObjectClear, which are far smaller in scale. This comparison seems somewhat unfair. Is the model's powerful ability to "perfectly remove objects without precise masking" due to the proposed RAA and OGA architecture, or simply because FLUX.1 carries a vast amount of prior world knowledge that provides a significant advantage? The paper lacks a baseline model based on FLUX.1 but without its core modules.
3. No failure cases are presented, and the scenarios in which the method fails (such as extreme occlusion, extremely small targets, complex textured backgrounds, and transparent objects) are not analyzed, making it difficult to determine the method's applicability.
4. In the core multilayer decomposition task, the authors only compared the results with CLD and Qwen-Image-Layered. Providing more state-of-the-art (SOTA) methods for comparison would make the results more convincing.
5. The header at the top of Figure 1 incorrectly spells "Original" as "Orignal".

---

> ### Author Rebuttal · Authors · 2026-03-30
>
> Thank you for your valuable and insightful comments.
>
> W1: Impact of inaccurate bboxes
>
> A1: Our method reduces token length by cropping the region, improving efficiency and performance. As a result, inaccurate bounding boxes can affect model performance.
> 1. We evaluate various noisy bbox settings (Reb-Table 2 in review rbJc-L1). The results show that mild perturbations, such as e2 (overly large boxes within 20%), e3 (≤5% offset), and e5 (≤5% inadequate), lead to negligible performance degradation. In contrast, more severe perturbations, such as e4 (5%–10% offset) and e6 (5%–10% inadequate), result in noticeable drops.
> 2. In practice, noisy inputs (e.g., text, boxes, masks, points) can be refined with SAM3 to obtain accurate regions for our model.
>
> W2: Unfair Backbone Comparison,  Architecture vs Backbone Contribution
>
> A2-1: The mainstream proprietary baselines (SDXL) target removal tasks, while our method addresses a more complex setting with box-guided separation, occlusion completion, and background recovery. We also compare with a FLUX-based method (OmniPaint), achieving comparable results, suggesting gains come from our design rather than model scale. Detailed results reported Reb-Table 1 (e3, e4, e6).
> A2-2: Table 4 in the paper analyzes removal capability. Table 4-(a), the flux-only baseline shows significantly weaker performance in both background recovery and foreground decomposition compared to the full model. Detailed results are provided in Reb-Table 1. Exp e1–e2 are Flux baselines, while e5–e6 are RevealLayer variants, showing clear improvements and highlighting the effectiveness of our design.
>
> Reb-Table 1.  Quantitative results  on object removal capability.
> | Exp | Method            | Type           | OBER-Test | & | & | & | RevealLayerBench | & | & | & |
> |-----|-------------------|----------------|-----------|-----------|-----------|-----------|------------------|------------------|------------------|------------------|
> |     |                   |                | PSNR↑     | SSIM↑     | LPIPS↓    | FID↓      | PSNR↑            | SSIM↑            | LPIPS↓           | FID↓             |
> | e1  | flux-inpaint      | inflated-mask  | 21.40     | 0.8513    | 0.1267    | 128.49    | 15.26            | 0.7716           | 0.3097           | 197.41           |
> | e2  | flux-fill         | inflated-mask  | 24.46     | 0.7957    | 0.1672    | 125.97    | 17.15            | 0.7917           | 0.2466           | 166.11           |
> | e3  | OmniPaint        | mask   | 29.05 | 0.8736    | 0.0521    | 20.70     | 24.83            | 0.8471           | 0.1374           | 54.22            |
> | e4  | OmniPaint        | inflated-mask  | 28.75     | 0.8702    | 0.0553    | 24.43     | 24.11            | 0.8421           | 0.1483           | 60.35            |
> | e5  | RevealLayer(flux-only)  | bbox   | 22.21     | 0.8844    | 0.0867    | 34.73     | 22.08            | 0.8357           | 0.1545           | 59.54            |
> | e6  | RevealLayer(full)  | bbox   | 30.16     | 0.9153    | 0.0694    | 25.62     | 25.53            | 0.8429           | 0.1483           | 53.81            |
>
> W3&L2: Failure Cases Analysis
>
> A3: We observe failure modes including misaligned boxes and complex scenarios (overlapping with rbJc-L1 and yzzp-W3).
> 1. Inaccurate boxes. Reb-Table 2(review rbJc-L1) shows severe misalignment (e4, e6) degrades performance.
> 2. Severe occlusion and complex transparent.Occlusions cause information loss, while transparent scenes lead to incomplete disentanglement; such cases are also hard to curate in training data.
> 3. Complex textured backgrounds. Failures may occur on dense or repetitive textures (e.g., grass, foliage) due to inherent ambiguity and multiple plausible reconstructions.
>
> In the revision, we will provide a more systematic analysis, to better characterize the applicability and limitations of our method.
>
> W4: Limited Comparisons in Multi-Layer Decomposition Tasks
>
> A4: Existing open-source datasets and methods for multi-layer decomposition are scarce. Recent works such as CLD(2025.11), QwenImageLayer(2025.12) are available, while RLD(ICLR 2026.02), OminiPSD(2025.12), and FromInpaintingToLayerDecomposition(2025.11) are not publicly released. Most existing methods are limited to single-foreground extraction or lack precise region control.
> To our knowledge, our method is the first to enable controllable multi-layer decomposition in general scenes. We will also release our model and dataset to support future research in the community.
>
> L1: High Inference Cost
>
> A5: Under the same experimental setting, the FLUX-only requires 93s, while RevealLayer takes 122s (reduced to 99s with FlashAttention). Detailed results are provided in Reb-Table 5(Review 44mu-Q3). For practical deployment, INT4 quantization and cache optimization can reduce latency to 57s with acceptable performance.
> Improving efficiency at both architectural and system levels remains an important direction for future work.

---

> > ### Author Rebuttal · Reviewer_Wt8J · 2026-04-03
> >
> > I still have some questions:
> > 1. The authors suggest using SAM3 to optimize noisy bounding boxes in practical applications. However, this completely contradicts the paper's core claims of being "end-to-end" and "bounding box-guided." Over-reliance on third-party models severely limits their usability in real-world scenarios (e.g., when external models are unavailable). A truly robust method should build fault tolerance internally, rather than outsourcing it to SAM3. I suggest the authors reconsider how to improve this capability.
> > 2. The authors claim that INT4 quantization reduces latency to 57s with "acceptable performance." This subjective qualitative description is too vague for academic evaluation. Please provide specific performance degradation metrics for the INT4 setting.
> > 3. Furthermore, RevealLayer (122s) is approximately 29 seconds slower than the FLUX-only baseline (93s). I want to know: is this additional overhead purely due to unoptimized engineering code, or is it an inevitable consequence of the extremely high algorithmic complexity introduced by your proposed RAA and OGA modules?
> >
> > At this time, I have decided to retain my initial rating.

---

> > > ### Author Response · Authors · 2026-04-04
> > >
> > > A1: RevealLayer is a bbox-guided framework for controllable layered decomposition. The role of the bounding box is to provide a simple yet effective spatial prior, enabling the model to reliably perform foreground decomposition, occlusion completion, and background recovery. Therefore, the key issue here is not whether the model can automatically correct arbitrarily poor bounding boxes, but whether it is sufficiently robust to box noise in real interactive scenarios. Our noisy-box experiments already answer this: mild perturbations, especially slightly oversized boxes, cause only limited performance degradation, whereas clearly shifted or overly tight boxes lead to more noticeable drops because they miss essential target regions or occlusion-related context. In practical use, a simple box dilation heuristic can further improve tolerance.
> > >
> > > At the same time, fully automatic layered decomposition without box guidance is a different problem setting. Methods such as Qwen-Image-Layered do not rely on explicit spatial priors, but as a result they offer weaker controllability and cannot benefit from region cropping for higher efficiency. Moreover, in our human evaluation, its layer decomposition success rate is only 57%, and its background recovery success rate is only 20%. Therefore, our goal is not automatic layer discovery without boxes, but rather accurate, efficient, and controllable layered decomposition under a lightweight box prior.
> > >
> > > A2: To evaluate the actual visual performance under the INT4 setting, we conducted a human evaluation comparing BF16 and INT4 inference. As detailed in Reb-Table 6, participants observed a small drop in perceptual quality for both background and foreground.
> > >
> > > A3: The additional runtime mainly comes from RAA. As shown in Reb-Table 5 (Review 44mu-Q3), the Flux-only baseline takes 93s, while RAA adds about 26s and OGA contributes only about 3s, resulting in 122s for the full model. This overhead arises from applying structured attention masks inside the transformer blocks.
> > >
> > > At the same time, this overhead is not entirely unavoidable. We further provide the optimized inference latency and performance after system-level acceleration. As shown in Reb-Table 6, enabling FlashAttention reduces the full-model latency from 122s to 99s, and applying INT4 quantization further reduces it to 57s, indicating that there is still substantial room for system-level optimization.
> > >
> > > Reb-Table 6. Human evaluation results comparing BF16 and INT4 inference.
> > >
> > > | **Method** | **Layer Count ↑** | **BG Quality ↑** | **FG Quality ↑** | **Time (s) ↓** |
> > > |------------|------------------:|-----------------:|-----------------:|---------------:|
> > > | BF16       | 99                | 85               | 90               | 99             |
> > > | INT4       | 99                | 80               | 88               | 57             |

---

### Decision · Program_Chairs · 2026-04-30

**Decision:**

Accept (regular)

**Comment:**

This paper works on controllable layered image decomposition which is an important task for user interactive image editing. The method is built on a FLUX/MM-DiT backbone with a fine-tuned transparent-image VAE. The paper also introduces the RevealLayer-100K dataset and a benchmark for the task.
This paper receives mixed reviews. One accept, one weak accept, two weak rejects. All reviewers recognize the novelty of the new dataset, conditioned that it will be published upon acceptance. Besides, some reviewers recognize the proposed setting (bounding box) is easier for users and harder for algorithms; and the authors demonstrated a reasonable solution for the new setting. On the negative side, concerns have been raised that the proposed method is computation heavy. And the experiments are insufficient.
In the end, there is no consensus among reviewers. The AC has checked the paper, reviews, rebuttals and authors’ correspondences. The AC finds the merit outweighs the demerit. Specifically, the dataset and the setting are novel, some technical issues can be solved by minor revision.  The AC therefore recommends weak acceptance.